



# Stress Characterization and Temporal Evolution of Borehole Failure at the Rittershoffen Geothermal Project

Jérôme Azzola[1], Benoît Valley[2], Jean Schmittbuhl[1], Albert Genter[3]

[1]Institut de Physique du Globe de Strasbourg/EOST, University of Strasbourg/CNRS, Strasbourg, France
[2]Center for Hydrogeology and Geothermics, University of Neuchâtel, Neuchâtel, Switzerland
[3]ÉS géothermie, Schiltigheim, France

*Correspondence to*: Jérôme Azzola (azzola@unistra.fr)

**Abstract.** In the Upper Rhine Graben, several innovative projects based on the Enhanced Geothermal System (EGS) technology exploit local deep fractured geothermal reservoirs. The principle underlying this technology consists of increasing the hydraulic performances of the natural fractures using different stimulation methods in order to circulate the natural brine with commercially flow rates. For this purpose, the knowledge of the *in-situ* stress state is of central importance to predict the response of the rock mass to the different stimulation programs. Here, we propose a characterization of the *in-situ* stress state from the analysis of Ultrasonic Borehole Imager (UBI) data acquired at different key moments of the reservoir development using a specific image correlation technique. This unique dataset has been obtained from the open hole sections of the two deep wells (GRT-1 and GRT2, ~2500m) at the geothermal site of Rittershoffen, France. We based our analysis on the geometry of breakouts and of drilling induced tension fractures (DITF). A transitional stress regime between strike-slip and normal faulting consistently with the neighbour site of Soultz-sous-Forêts is evidenced. The time lapse dataset enables to analyse both in time and space the evolution of the structures over two years after drilling. The image correlation approach developed for time lapse UBI images shows that breakouts extend along the borehole with time, widen (i.e. angular opening between the edges of the breakouts) but do not deepen (i.e. increase of the maximal radius of the breakouts). The breakout widening is explained by wellbore thermal equilibration. A significant stress rotation at depth is evidenced. It is shown to be controlled by a major fault zone and not by the sediment-basement interface. Our analysis does not reveal any significant change in the stress magnitude in the reservoir.



## 1 Introduction

Several deep geothermal projects located in the Upper Rhine Graben and based on the Enhanced Geothermal System (EGS) technology exploit local geothermal reservoirs, such as those located in Soultz-sous-Forêts or in Rittershoffen (Baujard et al., 2017; Genter et al., 2010). The principle underlying this technology consists of increasing the hydraulic performance of the reservoir through different types of simulations to achieve commercially interesting flow rates. The stimulation techniques are typically based on high pressure injection (hydraulic stimulation), cold water injection (thermal stimulation) or chemical injection (chemical stimulation). During the injections, a thermo-hydro-chemo-mechanical perturbation induces an increase in permeability due to the reactivation of existing structures or the generation of new ones (Cornet, 2015; Huenges & Ledru, 2011). The *in-situ* stress state is a key parameter controlling rock mass response during stimulation and is required to design stimulation strategies and forecast the response of the reservoir to varying injection schemes.

Despite its importance, the *in-situ* stress state is difficult to assess, particularly in situations where the rock mass is only accessible through a few deep boreholes. In such cases, the assessment of borehole walls using borehole logging imaging is a useful technique to provide information on the type, the orientation and the size of fractures or breakouts which are owed to the stress perturbations related to existence of the well (drilling and fluid boundary conditions). Subsequently, it gives useful constraints on the *in-situ* stress state surrounding the wellbore (Schmitt et al., 2012; Zoback et al., 2003). Borehole breakouts provide a indirect information on the stress orientation that it is difficult to extract in particular for robust quantitative stress magnitudes. Indeed, it relies on the choice of the failure model used to interpret borehole wall images. Indeed, the mechanisms that control the failure evolution of the borehole wall are not well understood both in space and time, and there is no consensus on the most appropriate failure criteria to be used. Parameterizing failure criteria is also a challenge since intact core material is often not available from deep boreholes. Finally, the set of images used to identify borehole failures is typically acquired a few days after drilling completion when it is unclear if the geometry has reached a new stationary state yet. The present analysis addresses these difficulties as we attempt to characterise the stress state at the Rittershoffen geothermal site (France).

We first present in this paper the geological and geodynamical context of the Rittershoffen geothermal site (France). We describe the borehole imaging data acquired in the GRT-1 and GRT-2 wells at the Rittershoffen geothermal project. We then proceed to a brief review of the methods used for UBI analyses with their underlying assumptions. We applied the methodology proposed by Schmitt et al. (2012) and Zoback et al. (2003) in order to assess the stress state at this site. To analyse the three successive images of the wellbore acquired up to two years after drilling completion, we developed an image processing method of the UBI data to compare in time the geometry of breakouts. We deduce from this study, the evolution of breakouts with time and evaluate its impact on our *in-situ* stress state assessment. We finally propose our best estimate of the *in-situ* stress state for the Rittershoffen site, both in orientation and magnitude.



## 2 Rittershoffen project context

The Rittershoffen geothermal project, also referred as the ECOGI Project is located near the village of Rittershoffen in North-Eastern France (Alsace). It is an EGS geothermal project initiated in 2011 (Baujard et al., 2015, 2017). The doublet has been drilled between Rittershoffen and Betschdorf, 6 km east of the Soultz-sous-Forêts geothermal project, in the Northern Alsace, France (Genter et al., 2010). The aim of the project is to deliver heat through a long pipeline loop to the "Roquette Frères" bio-refinery located 15 km apart. The power plant capacity is 24 MWth, intending to cover up to 25% of the client heat need. Figure 1 gives an overview of the project location and presents in the right insert the trajectory and completion of the two wells GRT-1 and GRT-2 that have been drilled (Baujard et al, 2017). GRT-1 was completed in December 2013. It was drilled to a depth of 2580 m (MD, depth measured along hole) corresponding to a vertical depth (TVD) of 2562 m. The well penetrates the crystalline basement at a depth of 2212 m MD and targets a local complex fault structure (Baujard et al., 2017; Lengliné et al., 2017; Vidal et al., 2016). The 8" 1/2 diameter open-hole section of the well starts at 1922 m MD. The borehole is almost vertical with a maximum deviation of 9° only. The first hydraulic tests concluded in an insufficient injectivity of the injection well GRT-1. Therefore, the well was stimulated in 2013, which resulted in a fivefold increase of the injectivity (Baujard et al., 2017). The target of the production well GRT-2 and its trajectory have been designed benefiting from the results of additional seismic profiles acquired in the meantime. GRT-2 targets the same fault structure but more than one kilometre away from GRT-1. Local complexities of the fault structure as 'in steps' geometry, has been observed *a-posteriori* from the micro-seismic monitoring during GRT-1 stimulation (Lengliné et al, 2017). The GRT-2 borehole was drilled in 2014 to a total depth of 3196 m MD (2708 m TVD) (Baujard et al., 2017). The granite basement is penetrated at a depth of 2493.5 m MD. The 8" 1/2 diameter open-hole section starts at a depth of 2120 m MD. This borehole is strongly deviated with a mean deviation of 37° over the interval of interest.

The left insert of Figure 1 shows more specifically the geological units penetrated by the deep boreholes of the geothermal sites in Rittershoffen and Soultz-sous-Forêts. It consists of sedimentary layers from the Cenozoic and Mesozoic that are overlaying a crystalline basement made of altered and fractured granitic rocks (Aichholzer et al., 2016). Natural fractures are well developed in the Vosges sandstones and Annweiler sandstones, as in the granitic basement. Oil and Gas exploration in the area led to a good knowledge of the regional sub-surface including measures of temperatures at depth. The unusual high geothermal gradient encountered in Soultz-sous-Forêts which is one of the largest described so far in the Upper Rhine graben, encouraged the development of the ECOGI project in this area (Baujard et al, 2017).

The geological context is characterized in the vicinity of the Soultz-sous-Forêts and Rittershoffen sites from numerous studies owing to the extended geophysical exploration in the region (Aichholzer et al., 2016; Cornet et al., 2007; Dezayes et al., 2005; Dorbath et al., 2010; Evans et al., 2009; Genter et al., 2010; Rummel, 1991; Rummel & Baumgartner, 1991). Given that GRT-1 and GRT-2 wells penetrate geologic units similar to those in Soultz-sous-Forêts, information from Soultz-sous-Forêts site can be used to better characterize the geological units through which the wells in Rittershoffen are drilled (Aichholzer et al., 2016; Vidal et al., 2016). It can be used in particular for the strength and mechanical characteristics



of these geological units which are poorly characterized at Rittershoffen site since no coring was made during drilling
(Heap et al., 2017; Kushnir et al., 2018; Villeneuve et al., 2018). The World Stress Map (WSM) released in 2008 also
compiles the information available on the present-day stress field of the Earth's crust in the vicinity and gives an overview of
the values and results which can be expected in Rittershoffen (Cornet et al., 2007; Heidbach et al., 2010;
Rummel & Baumgartner, 1991; Valley & Evans, 2007a). The data collected from WSM are presented in Figure 1 and
indicate that an orientation of the maximum principal stress close to N169°E and a normal to strike slip faulting regime are
expected for our study area.

## 3. Rittershoffen well data

### 3.1 GRT-1 data

Several extensive logging programs accompanied the drilling of wells GRT-1 and GRT-2. One was conducted in December
2012 in the open-hole section of GRT-1, few days after drilling (Vidal et al., 2016). UBI acquisitions were carried out
(Luthi, 2001). Figure 2 (b) shows the amplitude image acquired in 2012 in GRT-1 and Fig. 2 (c) displays the radius of the
borehole computed from the double transit time image. The well logging also included caliper, spectral gamma ray and
gamma-gamma acquisitions that enable an estimation of rock alteration and bulk density. The injectivity measured during
the first hydraulic test between December 30th, 2012 and January 1st, 2013 showed a low injectivity (Baujard et al., 2017).
To enhance the injectivity, the hydraulic connectivity between the well and the natural fracture network has been increased
through a multi-step reservoir development strategy. First a thermal stimulation of the well has been performed in April
2013. A cold fluid (12°C) was injected at a maximum rate of 25 L.s$^{-1}$ with a maximum wellhead pressure of 2.8 MPa. The
total injected volume was 4230 m$^3$. Second, a chemical stimulation followed in June 2013. Using open hole packers, a
glutamate-based biocide was injected in specific zones of the open hole section of GRT-1 (Baujard et al., 2017). Finally, a
hydraulic stimulation of the well has been performed in June 2013 with a large seismic monitoring at the surface
(Lengliné et al., 2017; Maurer et al., 2015). During these two last phases, a moderate volume injection, 4400 m$^3$ were
injected in the open hole. The hydraulic stimulation lasted during 30h, with a major phase of stepwise flow rates from 10L.s$^{-1}$
to 80 L.s$^{-1}$ (Baujard et al., 2017). As a result, the injectivity was improved fivefold due to this thermal, chemical and
hydraulic (TCH) stimulation program. Two other borehole imaging programs were conducted in December 2013 shortly
after stimulation of the well and significantly later in June 2015. The amplitude and travel time (or radius) images used in the
analysis are shown respectively in Fig. 2 (e) and Fig. 2 (f) for the logging program of 2013 and in Fig. 2 (h) and Fig. 2 (i) for
the logging program of 2015.
This time lapse UBI dataset, whose characteristics are summarized in Table 1, provides the essential information for the
present study as it enables to identify evidences of irreversible deformation and failure (natural and induced fractures,
breakouts, fault zones, damage zones, etc) along the borehole wall. Vidal et al. (2016) analysed the images acquired in
GRT 1 and identified fractured zones impacted by the TCH stimulation, without assessing the stress state and its evolution.





Hehn et al. (2016), whose measurements are discussed later in section 9.2, analysed the orientation of DIFTs in GRT-1 in the
granitic basement but also in the upper sedimentary layers, investigating the orientation of the stress field with depth.
We identify wellbore wall failure and use these observations to characterise the stress state in the reservoir, including its
evolution in time. Wellhead pressure measurements of the hydraulic stimulation are also used to estimate a lower bound of
the minimum horizontal stress *(Sh)*.
**3.2 GRT-2 data**
An extended logging program was also conducted in GRT-2, including repeated UBI borehole imaging (see Table 1). Figure
3(c) and 3(d) show respectively the amplitude image acquired in 2014, between 2404 m and 2412 m, and the radius image
acquired in 2015 between 2468 m and 2472 m, in GRT-2. No hydraulic stimulation was performed in this well since its
initial injectivity was sufficient (Baujard et al, 2017).
**4. Stress estimation methodology**
The approaches proposed by Zoback et al. (2003) and by Schmitt et al. (2012) are used to fully characterize the *in-situ* stress
field at the Rittershoffen geothermal project. In the following, the symbol $S$ refers to the total stress when $\sigma$ refers to the
effective stress (Jaeger & Cook, 2009). We suppose that one of the principal stresses of the *in-situ* stress tensor is vertical,
which is a common assumption. This hypothesis is justified by the first-order influence of gravity on the *in-situ* stress state,
although this assumption may not be valid locally. In the following, we denote the vertical principal stress, *Sv*. The
magnitude of the vertical stress *Sv* is obtained from the weight of the overburden. It is calculated by the integration of density
logs (see part 8.2). The two other principal stresses act horizontally: *SH*, the maximum horizontal stress and *Sh,* the
minimum horizontal stress. The magnitude of the minimum horizontal stress *Sh* is estimated from the wellhead pressure
measurements carried out during the hydraulic stimulation of GRT-1 and from the hydraulic tests performed in the reservoir
of Soultz-sous-Forêts (see part 8.3). The analysis of the borehole failures is evaluated using televiewer images data
(Zemanek et al., 1970; Zoback et al., 1985). The orientation and magnitude of *SH* is assessed using a failure condition at the
borehole wall.
**4.1 Wellbore stress concentration**
To express the stress concentration around the quasi-vertical borehole GRT-1 (maximum deviation is only of about 9°), we
assumed its shape to be a cylindrical hole, and used the well-known linear elastic solution, often referred to as the
Kirsch solution (Kirsch, 1898; Schmitt et al., 2012). For the deviated well GRT-2 where the plane strain approximation is
not valid anymore, we used a 3D solution taking into account its deviation (Schmitt et al. 2012). Note that we included in our
solution a thermal stress component that accounts for the thermal perturbation induced by the drilling process. This
component is detailed later in section 8.4. We used the formulation of the thermo-elastic stresses arising at a borehole given

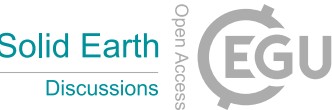



by Voight & Stephens (1982), also recalled in Schmitt et al. (2012). We computed the effective stress at the borehole wall
considering a hydrostatic pore pressure given by $Pp = \rho_f.g.z$, i.e. with the head level located at the surface. The fluid density
$\rho_f$, is taken as 1000 kg.m$^{-3}$ and the gravitational acceleration $g$, as 9.81 m$^2$.s$^{-1}$. $z$ is the vertical depth (TVD) in meter from
ground surface.

## 4.2 Failure criterion

At the scale of the surrounding of borehole (a few decameters), we assume a linear elastic, homogeneous and isotropic rock
behaviour prior to failure. When the maximum principal stress exceeds the compressive rock strength, rock fails in
compression (Jaeger & Cook, 2009). Failure at the borehole wall is assessed using the elastic stress concentration solutions
presented in part 4.1, combined with an adequate failure criterion. There is currently no consensus concerning the
appropriate failure criteria to assess wellbore wall strength. Since, in the case where the pore pressure and the internal
wellbore pressure are in equilibrium the radial effective stress at the borehole wall is equal to zero, a common assumption is
to consider that the Uniaxial Compressive Strength (UCS) is a good estimate of wellbore strength (Barton et al., 1988;
Zoback et al., 2003). Others suggest that the strength of borehole walls in low porosity brittle rocks could be less than the
UCS, because the failure could be controlled by extensile strains (Barton & Shen, 2018; Walton et al., 2015) or fluid
pressure penetration (Chang & Haimson, 2007). The presence of non-zero minimum principal stress and the strengthening
effect of the intermediate principal stress however suggest that the borehole wall strength should be larger than UCS
(Colmenares & Zoback, 2002; Haimson, 2006; Mogi, 1971). In view of this situation and because stress magnitudes
evaluation differs according to the criterion used in the analysis, we compared the estimates obtained using three commonly
used failure criteria in borehole breakouts analyses: 1) the Mohr-Coulomb criterion (Jaeger & Cook, 2009), 2) the Mogi-
Coulomb criterion (Zimmerman & Al-Ajmi, 2006) and 3) a true triaxial version of the Hoek-Brown criteria (Zhang et al.,
2010). The formulation of these criteria is given in the following equations (Eq. (1) to (3)) in the principal effective stress
space $\sigma_1 - \sigma_3$ for the Mohr-Coulomb criterion and in octahedral shear vs. mean stress space $\tau_{oct} - \sigma_m$ for the Mogi-Coulomb
and Hoek-Brown criteria:
$\qquad$ Mohr-Coulomb: $\sigma_1 \geq C_0 + q.\sigma_3$ $\hspace{6cm}$ (1)
$\qquad$ Mogi-Coulomb: $\tau_{oct} \geq a + b.\sigma_m$ $\hspace{6cm}$ (2)
$\qquad$ Hoek-Brown: $\frac{9}{2.C_0}.\tau_{oct}^2 + \frac{3}{2\sqrt{2}}.m_i.\tau_{oct} - m_i.\sigma_m \geq C_0$ $\hspace{4cm}$ (3)
Where $C_0$ is the uniaxial compressive strength and $q$ is a material constant that can be related to the internal friction angle, $\varphi$,
through $q = \left(\frac{\pi}{4} + \frac{\varphi}{2}\right)$. The octahedral shear stress is given by $\tau_{oct} = \sqrt{(\sigma_1 + \sigma_2)^2 + (\sigma_2 + \sigma_3)^2 + (\sigma_3 + \sigma_1)^2}$ and the mean
stress by $\sigma_m = \frac{\sigma_1 + \sigma_3}{2}$. The variables $a$ and $b$ in the Mogi-Coulomb criteria and $m_i$ in the Hoek-Brown criteria are parameters
that are related to the material friction and cohesion.



**5. Strength estimation**

Four simplified lithological categories have been used for the strength characterization of the rock at depth in the Rittershoffen reservoir. All the Triassic sandstones have been grouped in a single category. The granitic section has been separated in three categories according to the type and intensity of alteration. The simplified lithologic profile for GRT-1 and GRT-2 wells are indicated in Table 2. Considering the methodology used here, the relevance and accuracy of the stress characterization is highly conditioned by the values of the rock strength parameters and by the failure criterion chosen. In Rittershoffen, the drilling was performed exclusively in destructive mode and no sample is available to measure rock moduli and strength characteristics. Thereby, mechanical tests on core samples from the nearby Soultz-sous-Forêts site are used to characterize the rock properties (Rummel, 1991; Valley & Evans, 2006). Indeed, boreholes of both sites penetrate the similar lithological units and therefore using Soultz-sous-Forêts mechanical data for an application at the Rittershoffen site is considered acceptable. At Soultz-sous-Forêts site, EPS-1 borehole was continuously cored from 930 to 2227 m (Genter et al., 2010; Genter & Traineau, 1992, 1996) providing samples of the Sandstones in the Buntsandstein and in the crystalline basement. Some cores have also been obtained in the borehole GPK-1 from various depth sections and were analysed by Rummel (1992). For the Buntsandstein sandstones, because of the high variability of the rocks characteristics within this same geological unit and because only very few tests were performed on these sandstones, we rather used typical strength parameters (Hoek & Brown, 1997). The elastic and strength parameters used for our analyses are summarized in Table 2. The variability range given for elastic parameters, cohesion and UCS reflect natural rock heterogeneities and depict the variability in values encountered.

**6. Images processing and borehole failure identification**

Stress induced failures are identified and measured from acoustic borehole images. The confidence and accuracy of these determinations depend on the quality of the images. In the following, we describe the original data as well as the processing we applied to improve the quality and comparability of the images. We also explain how we measure borehole failure on these images and the limitations associated with these measurements.

**6.1 Quality of the acoustic televiewer images**

Several artefacts can deteriorate the quality of acoustic image data (Lofts & Bourke, 1999). The images acquired in Rittershoffen suffer from some of these limitations. The quality of the image depends of the tool specification, the acquisition parameters and logging conditions. All acoustic images at Rittershoffen were acquired by Schlumberger with their UBI (Ultrasonic Borehole Imager) tool. The tool and acquisition parameters were similar between each log, but not identical. For example, the GRT-1 log in 2013 was acquired using a smaller acquisition head (see the changes in transducer diameter detailed in Table 1. The acquisition resolution was the same for every log, i.e. 2° azimuthal resolution and 1 cm depth sampling step.




The 2012 log of GRT-1 has the best quality image of the entire suite. The image suffers of signal loss artefact
(Lofts & Bourke, 1999) in some limited sections, most commonly related to the presence of breakouts or major fracture
zones (Fig. 3 (a)).
The 2013 log of GRT-1 is of comparable quality than the 2012 log and suffers also of some limited signal loss artefacts. The
major issue with the image of GRT-1 acquired in 2013 is that the orientation module was not included in the tool string and
thus the image cannot be oriented with magnetometer data as it is usually done for this type of data.
The 2015 log of GRT-1 generally suffers from signal loss issues, not only in areas with major fracture zones and breakouts.
In the lower part of the log, wood grain artefact (Lofts & Bourke, 1999) is also observable (see Fig. 3 (b)). This is
particularly developed below 2431 m MD.
The quality of data of logs from GRT-2 is generally worse than the ones of GRT-1. This is due to the deviation of GRT-2
that makes wireline logging more difficult. The 2014 log of GRT-2 suffers from stick-slip artefacts on its entire length
(Fig. 3 (c)). The 2015 log in GRT-2 does not show any sign of stick-slip but present an erroneous borehole radius record
leading to an incorrect borehole geometry evaluation (Fig. 3 (d)).
Despite these difficulties, the images collected in the GRT-1 borehole are of excellent quality. Signal loss is the main
problem and it prevents to measure the depth in the radial direction of the breakout in some zones. Given the extent of the
artefacts highlighted in GRT-2, the measurements of the breakout parameters in this borehole are much more uncertain.

**6.2 Processing of the UBI images**

Prior the use of the images for assessing borehole failure, the images went through the following pre-processing steps:
1)  Transit time was converted to radius using the fluid velocity recorded during the probe trip down the borehole;
2)  Images were filtered to reduce noise;
3)  Digital image correlation was applied across the successive logs in order to correct the image misalignment both in
azimuth and depth.
The borehole radius was computed from the transit time following Luthi (2001):
$$r = \frac{t_{twt} \cdot v_m}{2} + d \qquad (4)$$
with $t_{twt}$ the two-way travel time, $v_m$ the acoustic wave velocity in the drilling mud, and $d$ the logging tool radius. Images are
filtered using a selective despiking algorithm implemented in WellCad™ using a cut-off high level (75%) and a cut-off low
level (25%) in a 3x3 pixels window. The goal of this process is to replace outliers by cut-off values when the radius exceeds
the cut-off high or low level. Finally, digital image correlation was used to insure proper alignment of the UBI images. This
was required for the GRT-1 2013 image because this image was not oriented with a magnetometer/accelerometer tool. The
process was also applied to the 2015 GRT-1 data to facilitate comparison between images. For this purpose, we developed a
technique based on a Particle Image Velocimetry (PIV) method (Thielicke & Stamhuis, 2014) that relies on optical image
correlation but being applied to travel time UBI images. This image alignment process is illustrated in Fig. 4. Figure 4 (a)



shows as example the "correlation box" in the travel time UBI image of reference - i.e. 2012 in this case – and the corresponding one in the image to compare with - i.e. the image of 2013 – which it is shifted of a given displacement vector ($dX$, $dY$) within the "search box". The cross-correlation function, which is a measure of the similarity between the thumbnails, is computed between the correlation boxes for each displacement vector ($dX$, $dY$). Right panel of Figure 4 (a) shows a map of the cross-correlation function computed for every displacement vector in a given search box. The two-dimensional cross-correlation function is an operator acting on two intensity functions $s(X,Y)$ and $r(X,Y)$, defined as a norm of the colour levels at each position of each thumbnail. $C_{sr}$ is defined at a position $(X,Y)$ and for a shift $(dX, dY)$ by Eq. (5):

$$C_{sr}(dX, dY) = s(X,Y) \otimes r(X,Y) = \iint_{-\infty}^{+\infty} s(X,Y) r * (X - dX, Y - dY) dX dY \tag{5}$$

The position $(dX, dY)$ within the "search box" with the highest cross correlation correspond to the best alignment (see Fig. 4 (a)). The operation is repeated along the image for each position of the search box. Importantly, the correlation box is taken with an anisotropic shape to account for the rigid rotation of the UBI tool and the linear property of the acoustic camera. The size of the correlation box is 180 x 20 pixels. This configuration is appropriate to identify principally the azimuthal offset while it is less sensitive to the depth mismatch. We investigated offset up to 180 pixels horizontally corresponding for our 2° resolution to a complete 360° rotation. We considered vertical offset of ± 10 pixels corresponding to offsets of about ± 10 cm. Figure 4 (b) gives an example of image realignment and shows the efficiency of the process. This correlation process allows to align finely the successive images and thus to study the borehole shape evolution with time more accurately.

### 6.3 Determination of the borehole failure

For GRT-1, the breakouts have been determined through a visual analysis of borehole sections computed every 20 cm from 1926 m to 2568 m (MD) from the double transit time data. The borehole sections are computed by stacking (averaging using the median) the data collected every 1 cm over 20 cm borehole interval (with no overlap between two successive sections). The median is thus used because it is less sensitive to extreme values than the mean and thus is efficient at removing local noise from the data. Prior to determining breakout geometrical parameters, the actual borehole center is determined by adjusting the best fitted ellipse to the borehole section. This process corrects for eventual logging probe decentralisation. For each section presenting the characteristic elongated shape of breakouts due to stress induced failure, the azimuthal position of the edges and the center of each limb is determined by visual inspection. Figure 5 gives examples of such determination to depict the process. The breakout edges are defined as the location where the wellbore section departs from a quasi-circular section adjusted by the best fitted ellipse. As it can be seen in Figure 5, this typically spans an azimuthal range much broader than the low amplitude reflections visible as dark bands on the amplitude images and justifies the choice to use the double transit time data. The positions of the breakout edges are not easy to determine in a systematic and indisputable manner, and





a significant uncertainty is associated with these measurements. Related to this issue, it is not possible to determine on the
images what azimuthal range of the wellbore is enlarged by purely stress redistribution processes and what part is enlarged
subsequently by the effects of drill strings wear. These uncertainties about the physical process controlling the enlargement
of the breakout could limit the comparisons between the three successive logs acquired in GRT-1. Breakout measurements
were thus performed on all three images concomitantly and consistently. We controlled for example that within a tolerance
dictated by the uncertainties of the measurements, the width of breakouts only remains identical or increases: no decrease in
width is measured between successive logs.
Figure 2 (d), (g) and (j) summarize all the measurements of the breakout's geometry performed in GRT-1, for the images
acquired in 2012, 2013 and 2015. Black dots indicate the azimuth at which the radius of the breakout is maximum and red
bars link the azimuthal position of the breakout edges used to compute the width of the breakouts. Given the difficulty of
measuring breakouts as discussed previously (i.e. artefacts affecting the images, disputable positions of the breakout edges),
a confidence ranking has been established for each breakout. This confidence level is presented in Fig. 2 (k). From the
geometry of the breakouts, we compute the breakout widths which are obtained from the breakout edge azimuths. The
deepest point of the breakout is used to determine the enlargement radius. In some situations, signal loss issues prevent the
determination of the enlargement radius, as it is shown in Fig. 5 for the image of GRT-1 acquired in 2015. The measured
width (black dots, in degree) and enlargement radius (red dots, in mm) are determined from the GRT-1 data set acquired in
2012 and presented in Fig. 2 (l).
Drilling Induced Tension Fractures (DITFs) are also identified from the GRT-1 borehole images using the same procedure as
for the breakout determination. For example, clear DITFs are evidenced in the amplitude image from 2395 m to 2400 m in
GRT-1 and presented in Fig. 6. Green crosses show the azimuth of the DITFs that is measured in GRT-1 every 20 cm. Blue
dots in Fig. 2 (d), (g) and (j) summarize the azimuth of the DITFs measured in GRT-1, respectively in 2012, 2013 and 2015.
Given the poor quality of the double transit time images acquired in GRT-2, less focus has been given to the analysis of the
borehole failure in this well. The data set consists of the acquisitions made in 2014 after completion of the borehole and in
2015. The investigated depths vary from the 2014 to the 2015 dataset. It is from 1950 m (Vertical Depth – 2220 m MD)
down to 2125 m (TVD – 2440 m MD) in 2014 when it is down to 2160 m (TVD – 2480 m MD) in 2015. The well is
strongly deviated. The concentration of stresses within the borehole wall is expressed under the assumption of a constant
deviation of 37° and measurements carried out as a function of the True Vertical Depth, to be comparable with the results
obtained in GRT-1 which is considered as vertical. Borehole sections are computed every 50 cm. To this end, borehole
sections are stacked using the data collected every 1 cm over 50 cm borehole interval, all along the transit time image. As for
GRT-1, the actual borehole centre is determined by adjusting a best fitted ellipse to the borehole section. Breakouts are
analysed by visual analysis in a same manner as for GRT-1 data. The difficulties encountered with the identification of
breakout geometry are more pronounced for images acquired in GRT-2 as artefacts are more developed. The deviation of
this well results on pronounced stick-slip effects. For a more accurate comparison between the measurements carried out on



the images acquired in 2014 and 2015, measurements are performed for the two images concomitantly. No DITFs are identified on the GRT-2 borehole images.

## 7. Analyses of temporal borehole failure evolution

The characterization of the stress tensor derived from the analysis of borehole failures typically relies on a single borehole image data set. From this snapshot in time, stresses are estimated while information on the evolution of breakout shape in time is not available. Interestingly, for the ECOGI project, the acquisition of three successive image logs allows to study this evolution. Here, the time evolution of breakouts, referred as breakout development, is analysed to characterize the time evolution of the borehole failure. A common hypothesis concerning borehole breakout evolution is that their width remains stable and is controlled by the stress state around the well at the initial rupture time. Progressive failure is supposed to lead however to breakout deepening until a stable profile is reached (Zoback et al., 2003).

An example of a time-lapse comparison of breakout shapes is presented in Fig. 7. Images of GRT-1 from 2012, 2013 and 2015 show a clear breakout at a depth of about 2126 m in the "couches de Trifels" in the Buntsandstein. Breakouts can present three types of evolution:

1) They can develop along the well, corresponding to an increase in the vertical length of breakouts. We refer to this process as *breakout extension*;

2) They can widen, corresponding to an apparent opening between the edges of the breakouts. We refer to this process as *breakout widening*;

3) They can deepen, corresponding to an increase of the maximal radius of the breakout (or "depth" of the breakout) measured in the borehole cross section at a given depth. We refer to this process as *breakout deepening*.

Figure 7 shows the evolution from 2012 to 2015 of the breakouts, at 2125.6 m. Failure did not occur in 2012 while breakouts are visible in 2013 and 2015. When superposing the 2013/2015 borehole sections, no change in breakout shape is highlighted for the west limb although a slight widening is visible on the east limb. Possible deepening of the east limb is occulted by signal loss issues. The borehole section computed at 2126.2 m shows on the contrary, no modification of the breakout shape from 2012 to 2015 in GRT-1.

Development of borehole failures depends also on the lithology. Breakout extension (longitudinal failure development) is quite common in the Buntsandstein while it is very limited in the basement granites, which is highlighted in Fig. 8. The evolution occurs exclusively between the 2012/2013 data set while no longitudinal extension occurs during 2013 and 2015. In 2012, a total breakout length of 404 m is observed. It increases to 504 m in 2013 and then remains stable in 2015 with a length of 506 m. There is no clear evolution of DITFs along the GRT-1 well despite the hydraulic and thermal stimulation performed between 2012 and 2013.

Figure 9 shows an increase of breakout width. We first compare the data acquired in 2012 and in 2013. 73% of the change of width is within an interval -10° / +10°, i.e. within our measurement uncertainty. For these breakouts no changes of width can





be highlighted within our level of uncertainty. However, for 27% of our data, we observe an increase of width larger than
10°. This is reflected by the long tail (with values higher than 10°) of the histogram computed from the width of breakouts
(see Fig. 9 (c)). The widening of these breakouts is undisputable. When comparing the data acquired in 2013 and in 2015,
very little changes are observed. Indeed, most of the measured changes remain below our uncertainty level of ±10° (red
histogram on Fig. 9 (c)).
The evolution of the maximum radial extension (breakout deepening) of the breakout measured in the borehole cross
sections is presented in Fig. 10. This parameter is more delicate to track because of signal loss issues (see for example
Fig. 3 (a)). In our analysis, we filtered out obvious incorrect depth measurements related to these artefacts, i.e. when the
computed radius from transit time image is clearly shorter than the drill bit radius. For both time intervals (2012-13 and
2013-15), the change in the depth of the breakout is symmetrically distributed around 0 mm and spans a variability of about
±15 mm. We interpret this distribution as an indication that if any deepening occurred, it remained within our uncertainty
level. Our data analysis does not enable to conclude in a general deepening of the breakouts.

## 8. Stress characterization

We propose in this section a complete stress characterization at different periods in both the GRT-1 and GRT-2 wells,
including a thermal history and thermal stress analyses and discuss the impact of breakout widening in time on stress
estimation. To that purpose, we first determine the orientation of the stress tensor. We then detail how we estimate the
minimum horizontal stress component $Sh$, the vertical stress component $Sv$ and the thermal component. Finally, we propose
an estimation of the maximum horizontal stress component $SH$ from the measurement of the width of breakouts.

### 8.1 Maximum horizontal stress $SH$ orientation

The orientations of breakouts and DITFs are a direct measure of the principal stress directions in a plane perpendicular to the
well. As discussed previously, we assume that $Sv$ is in-overall vertical which is a common hypothesis in such an approach
and is justified by the first-order effect of gravity on *in-situ* stresses. In GRT-1 which is considered as vertical, DITFs are
aligned with the direction of the maximum horizontal stress ($SH$) and breakouts are aligned with the direction of minimum
horizontal stress ($Sh$).
Figures 2 (d), (h) and (i) show the orientation of breakouts (black dots) and DITFs (blue dots) measured in GRT-1. The
measurements are compiled in Fig. 11 as circular histograms. We chose to only analyse data from the images acquired in
2012 and in 2015. Indeed, data acquired in 2013 were obtained without orientation since the device was not functioning
correctly and are reoriented with respect to the 2012 data. Subsequently, the measurements carried out in the 2013 image do
not bring additional constraints in terms of stress orientation.
In the Buntsandstein sediments, the failure orientation is stable and indicates that the principle stress $SH$ is oriented
N15° ±19° (one circular standard deviation). The same failure orientation persists in the upper section of the granite down to





about 2270 m. Below this depth borehole failure orientation is much more variable as it seems to be influenced by the
presence of major fault zones crossing the GRT-1 borehole at a depth of 2368 m (MD) (Vidal et al., 2016). Below 2420 m,
which is the deepest large structure visible on the GRT-1 borehole image, the failure orientation indicates that $SH$ is oriented
165° ±14°. This is significantly different from the orientation in the sediments with a 30° counter-clockwise rotation. Such
differences in orientation with lithologies have already been noticed by Hehn et al. (2016) from the analysis of the
orientation of drilling induced fractures observed on borehole acoustic logs acquired in GRT-1. The orientation of $SH$
proposed by Hehn et al. (2016), i.e. globally N155°E in the basement and N20°E in the sedimentary layer, is consistent with
our measurements.
The geological study of the cuttings from the drilling of GRT-1 and GRT-2 enabled to determine the rock density profile in
both wells (Aichholzer et al., 2016). Thanks to this analysis, we estimate the mean density of each lithological layer. Table 3
shows the rock volumetric mass density as a function of the vertical depth (TVD). The magnitude of the vertical component
$Sv$ at depth is computed accordingly by integrating the volumetric mass density profile and a linear regression is fitted to the
measurements obtained from surface. In the following, the vertical component $Sv$ is computed from a linear trend expressed
as a function of vertical depth (TVD) $z$:
$$Sv\ [MPa] = 0.024\ .\ z\ [m]\ -\ (0,83) \tag{6}$$

### 8.3 Minimum horizontal stress Sh

We take the first order assumption that the minimum horizontal stress $Sh$ varies linearly with depth. Usually, the minimum
horizontal stress $Sh$ is estimated at depth from hydrofracture tests, (i.e. Valley & Evans (2007)) but this was not done at
Rittershoffen site. If the data available for the ECOGI project limit our analysis of the minimum stress component, numerous
injection tests were conducted in Soultz-sous-Forêts. We present in Fig. 12 their main trends. The injection tests performed
in the deep wells (GPK-1, GPK-2 and GPK-3 or EPS-1) of Soultz-sous-Forêts (Cornet et al., 2007; Valley & Evans, 2007b)
give important constraints for the minimum horizontal stress $Sh$ at the Rittershoffen site for large depths. In addition, the
study of Rummel & Baumgartner (1991) provides estimates at shallow depth. We compute the horizontal minimum stress $Sh$
as a function of the true vertical depth (TVD) $z$ from the linear trend proposed by Cornet et al. (2007) for the site of Soultz-
sous-Forêts (Figure 15):
$$Sh[MPa] =\ 0.015.\,z\,[m]-\ 7.3 \tag{7}$$

In order to check the applicability of the linear trend inferred from acquisitions in Soultz-sous-Forêts to the Rittershoffen
site, we estimated a lower bound of the minimum horizontal stress $Sh$ at 1913 m in Rittershoffen from the measure of the
wellhead pressure during the hydraulic stimulation of GRT-1 (Baujard et al., 2017). Figure 13 shows that the variation of
wellhead pressure with the flow is slower during the high rate hydraulic stimulation (above 40 L.s[-1]) than during the low rate
hydraulic stimulation (below 40 L.s[-1]). This change in behavior is interpreted as the beginning of a pressure capping resulting



from fractures reactivation. Hydraulic stimulation typically increases pore pressure which reduces the effective stress until
pressure equals *Sh* in magnitude. In theory, an increase of pressure could activate new fractures which results in the capping
of the recorded pressure: in such a case, minimum horizontal stress is inferred at depth from the maximum pressure achieved
during such a test. Meanwhile, other processes (shearing of existing weak fractures for example) could possibly result in the
capping of pressure for lower pressure values. From Figure 13, we assume that wellhead pressure caps at 22.6 MPa at 1913m
(TMD) for a flow rate 80 L.s$^{-1}$ (Fig. 12). It provides a lower bound to constrain the minimum horizontal stress *Sh* at depth,
which is compared to the Soultz-sous-Forêts trends in Fig. 13 and shows the consistency of the linear trend used in our
analysis.

**8.4 Thermal stresses**

The cooling of the well imposed during drilling, results in a thermal stress contribution. Accordingly, the characterization of
the stress tensor necessitates to include a thermal stress analysis which requires a good knowledge of the thermal history of
the well. We define the thermal contributions in the stress concentration at the borehole wall as: $\sigma^{\Delta T} r$ , $\sigma^{\Delta T}{}_z$ and
$\sigma^{\Delta T}{}_\theta$ respectively the radial, vertical and tangential components. The thermal stresses resulting from the temperature
difference, *Δt*, between the borehole wall and the so called ambient temperature, i.e. the initial temperature at that depth
before the drilling phase or the temperature at a significant distance from the borehole (not influenced by the borehole
perturbation), are expressed from Voight & Stephens (1982). These authors adapted the thermo-elastic solutions proposed by
Ritchie & Sakakura (1956) for a hollow cylinder to study the stress concentrations at the borehole wall due to the application
of a temperature difference. The radial component is null, and the tangential component is expressed as:
$$\sigma^{\Delta T}{}_\theta = \sigma^{\Delta T}{}_z = \alpha.\,E.\,\frac{\Delta T}{(1-\nu)} \tag{8}$$
where $\alpha$ is the thermal expansion, *E*, the Young modulus and **ν**, the Poisson ratio. Figure 14 (green curve) presents the
temperature log acquired in 2015 in GRT-1 (Baujard et al, 2017). It is plotted along with the temperature log acquired in
2013 (red curve). The comparison shows that temperature is close to be stable during that period in GRT-1. As a result, the
temperature log acquired in 2015 in GRT-1 is used as an estimate of the ambient temperature since it is considered as in
equilibrium with the reservoir. Temperature at the borehole walls at drilling completion is best estimated from the
temperature log acquired four days after drilling competition. The temperature log is presented in Fig. 14 (blue curve) and
the difference in temperature *Δt* computed from these logs is presented in the right panel of Fig. 14. Interestingly, these
temperature logs show a clear anomaly at 2360m where the wells are crossing the main fault zone associated to a major
permeable structure that controls two third of the total flow during flow tests (Baujard et al., 2017).



## 8.5 Maximum horizontal stress SH magnitude

The determination of the azimuthal position of the breakout's edges and of their width from the analysis of the UBI images acquired in GRT-1 and GRT-2 enables to estimate the maximum horizontal stress *SH*, and to evaluate its evolution with depth and time. Here, we present the results of our inversion, at multiple dates in GRT-1 and GRT-2.

In GRT-1, we obtain for each UBI log (in 2012, 2013 and 2015), three estimates of the magnitude of *SH*, according to the failure criterion. Figure 15 shows estimates of the magnitude of *SH*. The maximum horizontal stress *SH* in GRT-1 is presented for the 2013 UBI log as a function of the true vertical depth (TVD), along with the *Sh* and *Sv* obtained previously (Eqs. (6) and (7)). The horizontal error bars are calculated from the uncertainty on the elastic parameters, on the *Sh* and *Sv* estimates and on the measurements of the width of the breakouts. The uncertainty *ΔSH* is obtained by integration, taking into account the uncertainty *Δxi* on each variable $x_i$ involved in the estimation of *SH,* i.e the strength parameters, the *Sh* and *Sv* trends and the width of the breakouts:

$$\Delta f = \sum_i \left| \frac{\partial f}{\partial xi} \right| . \Delta xi \tag{9}$$

Figure 15 shows that the *SH* magnitudes vary significantly with the failure criterion. In particular, it shows that the *SH* stresses computed using a criterion that considers the strengthening effect of the intermediate principal stress (i.e. in Mogi-Coulomb or Hoek Brown) are higher than those calculated from a criterion that considers only the minimum and maximum principal stresses (i.e. in Mohr-Coulomb).

To choose the criterion that best describes the failure in the borehole, we use the approach proposed by Zoback et al. (2003) to display the stress state estimates presented in Fig. 15 in the stress polygon whose circumference is defined by a purely frictional, critically-stressed Earth crust. For this purpose, we suppose that crustal strength is limited by a Coulomb friction criterion with a friction coefficient *μ* = 1. We considered a depth of 2500 m to evaluate the vertical stress and assumed a hydrostatic pore pressure. The possible stress states from 2013 UBI images, are shown in Fig. 16 in a normalized *SH* vs *Sh* space. Because 2500 m is an upper boundary for the investigated depths in our study, the circumference of the polygon sets a maximum value for the maximum and minimum horizontal stresses *SH* and *Sh*. The stresses are normalized by the vertical stress magnitude *Sv* to facilitate the comparison. The maximum principal stresses *SH* measured using both our parametrized Hoek-Brown and Mogi-Coulomb criteria (blue and black dots) exceed the polygon boundaries. With our selection of parameters, the Mohr-Coulomb criterion was therefore retained as the most suitable for characterizing rock failure in our study. The same conclusion was drawn by Valley & Evans (2015) in Basel.

For GRT-2, we calculated the *SH* magnitudes using only the Mohr-Coulomb criterion retained in the previous analysis. GRT-2 is highly deviated and the well has been imaged in 2014 and 2015. The deviation is constant in the section of interest (i.e. the open hole): 37° N355°E. *SH* stresses are shown as a function of the vertical depth (TVD) in Fig. 17 with the according error bars and plotted along with the *Sh* and *Sv* trends in GRT-2.



The impact of breakout widening on stress estimation can be evaluated from our time-lapse characterization of the stress
tensor in GRT-1 and GRT-2. For GRT-2, Fig. 17 shows that *SH* magnitude changes are limited between 2014 and 2015,
given the uncertainty on the estimates. Figure 18 compares the *SH* stresses estimated using the Mohr-Coulomb criterion at
different dates in both GRT-1 and GRT-2 wells. The systematic shift observed between the estimates in both wells suggest
that the lower stresses estimated in the deviated well lead to a borehole wall stress concentration closer to the failure
condition than in the vertical well. Figure 18 evidences a time evolution of the *SH* stress estimates in GRT-1. Panel b.
quantifies the differences in *SH* stress between 2012 and 2015 in GRT-1 in a 1 MPa bins histogram. The confidence in the
time-evolution, is discussed in the next section considering the error on *SH*.

## 9. Discussion

The data set from the Rittershoffen geothermal project and our analyses allow us to discuss both the evolution over time and
with depth of the observed borehole failures. The impact of these evolutions on our ability to estimate stress magnitude from
borehole failure indicators is important.

### 9.1 Evolution of breakout geometry with time

Our analysis of the evolution of the breakouts geometry with time proves a development of breakouts along the well GRT-1
during the first year after drilling (Fig. 8). Indeed, we highlighted that sections without breakouts in 2012, four days after
drilling, present characteristic breakouts in 2013 and 2015, respectively one year and 2.5 years after drilling. We also
observe numerous lengths increases of the 2012 existing breakouts with time in particular in the Buntsandstein. The
difficulty is to link this evolution with time with a specific process: time-dependant rheology of the rock (i.e. creep) or the
effects of one of the stimulations, thermal, chemical or hydraulic. Moreover, the 2012 data were acquired at a period during
which the thermal perturbations due to the drilling operations were still present. The data they are compared with have been
collected in 2013 or 2015, after hydraulic, thermal and chemical stimulations of the well. As a result, the observed changes
could have taken place during the thermal equilibrium of borehole after drilling or during the simulations operations, i.e.
directly after drilling or later.
The conclusion brought by our time-evolution analysis of the breakout's geometry contradicts the usual assumption that
breakouts deepen (i.e. an increase in the maximum radius measured in the borehole cross sections) but do not widen (i.e. an
opening between the edges of the breakouts) with time (Zoback et al. 2003). However, the statistical approach applied in our
study along the open-hole of the well GRT-1 must be interpreted with caution. Even if we propose a systematic analysis of a
time-evolutive dataset, signal loss artefacts prevent an accurate measurement of borehole radius at some depths. It limits
locally our ability to reliably estimate the depth of the breakout, i.e. the extension of the breakout in the radial direction.
Given this limitation, we do not totally exclude that breakouts could have deepen with time. Our breakout width evaluation
is also affected by uncertainty: the deviation from the nominal cylindrical borehole geometry of the borehole adds



complexity to the measurements made considering the disputable positions of breakout edges. Meanwhile, we mitigated this difficulty by proposing a systematic analysis of all dataset to ensure a more consistent measurement and by attributing an uncertainty level on these values. Our study is thus more conclusive concerning this geometric parameter given that measured changes exceed our uncertainty level.

The widening observed in our data set can be explained by the process of thermal stress dissipation. Indeed, the 30 to 35°C of cooling observed at the time of the 2012 logging, are dissipated by the time of the 2013 logging (see Fig. 14). Assuming thermo-elastic properties of the material, the thermal hoop stresses implied by the cooling reaches -17 to -20 MPa (Eq. (8)). This will be sufficient to explain the change in breakout width without including additional time-dependent failure processes.

## 9.2 Evolution of breakout geometry with depth

The development of breakouts depends on the rock rheology and subsequently on the lithology. For our data set, breakouts are more numerous and extended in the sedimentary cover than in the granitic basement (Fig. 2). Moreover, their development is more pronounced in the sedimentary cover when they develop with time, vertically along the well (Fig. 8). Both observations are consistent with the fact that the sediments have on average a lower strength compared to the granitic rocks (Evans et al., 2009; Heap et al., 2019; Kushnir et al., 2018), i.e. conditions are closer to failure in the sediments.

Another important aspect of the variation of breakout geometry with depth is the evolution of their mean orientation. From the combined measure of the azimuth of maximum radial extension of the breakouts (BOs) and of the azimuth of Drilling Induced Tensile Fractures (DITFs), we analyse in Figure 11 the evolution with depth of the orientation of the maximum principal stress $SH$. The measurements are repeated for the images acquired in GRT-1, in 2012 and in 2015. The consistency between the orientation of our data between the 2012 and the 2015 data set (the 2013 data set was not oriented) builds confidence in the reliability of these indicators.

Figure 11 suggests that the orientation measured in the granitic layers below 2420m in Rittershoffen is consistent with the measurements carried out in the basement of Soultz-sous-Forêts (Valley & Evans, 2007b) and tends to reach the regional orientation. The red line in Fig. 11 is a moving average of the orientation data. It is computed over a 20 m window in depth. The measurement is carried out only if 50 individual measurements or more are present in the averaging window. It shows that the orientation of the maximum principal stress $SH$ varies in the studied section. Another important aspect of Figure 11 is the significant rotation of 30° from NNW to NNE highlighted between the bottom and the top of our analysed section. Such rotation with depth has already been evidenced in the Upper Rhine graben area in the Basel geothermal boreholes (Valley & Evans, 2009), in potash mines (Cornet & Röckel, 2012) and at the neighbouring geothermal site of Soultz-sous-Forêts (Valley & Evans, 2007b). Hehn et al. (2016) have also evidenced local stress rotations in the sedimentary section of GRT-1 up to the upper Triassic (Keuper) from the analyses of DITFs. The orientation measured here above the limit set close to 2400m MD (Fig. 11), is also consistent with the measurements of Hehn et al. (2016). They interpreted these variations to be related to mechanical contrasts between stiffer and softer rock layers. The particularity of the measurements proposed in Fig. 11 is that the orientation of the maximum principal stress $SH$ deviates from the regional trend within the



granitic basement, while the measurement in the upper basement aligns with the orientation of the sedimentary cover
(Fig. 11). The presence of a major fault crossing the GRT-1 borehole at a depth of 2368 m MD (Vidal et al., 2016) could be
the explanation of this rotation. The location of the observed stress rotation, i.e. in the basement and around 50m above the
major fault zone, doesn't assume that it is related here to the stiffness contrast or decoupling between the sedimentary cover
and the underlying basement as typically assumed, but rather to the presence of a neighbouring major fault zone.
Considering a high dipping fault geometry for this fault zone, it suggests that the geothermal well tangents the fault zone
explaining why breakouts are observed below but also above the major drain of the fault zone located at 2368 m (Fig 11).
Moreover, it was clearly demonstrated based on continuous granite core analyses at Soultz, that fault zone could have a
significant thickness due to the presence of a damaged zone characterized by an intense hydrothermal alteration
(Genter et al., 2010). Therefore, the absence of breakouts visible in the altered granitic section located just above the main
fault drain and the anticipated rotation of the stress field at some distance in the hanging wall and the footwall of the fault
zone confirm its major mechanical influence.

### 9.3 Evaluation of stress magnitude from breakout width

Our study shows the sensitivity to the failure criterion. Our analyses suggest that the Mogi-Coulomb and Hoek-Brown
criteria tend to overestimate borehole wall strength because they lead to stress estimates that violate frictional strength limit
of the crust (Fig. 16) while the Mohr-Coulomb strength model leads to acceptable results. This conclusion is however
dependent of the detailed parameterization of the failure criterion which is in Rittershoffen supported by sparse data. The
rock strength is among the main parameters that impact the stress magnitude assessment. At the Rittershoffen project, we
have no access to direct strength measurements since no cores were collected. We rely on measurement at the neighbouring
Soultz-sous-Forêts site where cores are available. However, even at Soultz-sous-Forêts, a systematic characterization of the
rock strength of the various lithologies is not achievable, particularly for the sediments.
In addition to the uncertainty on the strength parameterization, the uncertainty on width determination and the evolution of
width with time further impact the stress estimation. In the case of the GRT-1, significant changes occur between the 2012
data set (prior to reservoir stimulation operations) and the 2013-15 data sets (after stimulation). Panel (b) of Figure 18 shows
that the changes in the $SH$ stresses between 2012 and 2015 in GRT-1 are larger than our measurement uncertainty for 15% of
the measurements and are showing principally stress increases. This change can be fully explained by the thermal
equilibration of the well. The uncertainty on our data doesn't allow to relate stress changes to the reservoir stimulation
operations.

### 9.4 Stresses magnitude evolution with depth

Stresses estimated in GRT-1 and GRT-2 suggest that $SH$ is generally close to the vertical principal stresses $Sv$ consistently
with a transitional regime between both strike-slip regime and normal faulting regime (Anderson, 1951). This is consistent
with the stress characterization of the neighbouring site of Soultz-sous-Forêts, where measurements have highlighted a





normal faulting regime in the top granitic layers evolving into a strike slip regime more in depth. The uncertainty about our measurements and about the strength parameterization does not allow, however, for a decision on the faulting regime and its evolution with depth in Rittershoffen. A step in *SH* magnitude is visible on our estimate in Fig. 18 at large depth (below 2250m). This step occurs at the interface sediment basement and could be explained by the effect of stiffness contrast between lithologies (Corkum et al., 2018).

## 10. Conclusion

Thanks to the repeated UBI logging of the geothermal wells GRT-1 and GRT-2 in Rittershoffen (France), this study focuses on the analysis of the evolution with time and depth of the borehole breakouts. The following conclusions are drawn:

(i)  Clear evidences of time evolution of the breakout exist in particular in the sedimentary cover.

(ii)  The evolution in time of the vertical length and the horizontal width of the breakouts are measured benefiting from the development of a UBI image correlation technique It is discussed in the limit of the estimated uncertainties. The vertical length of the breakouts is shown to increase with time. No variation in the depth of the breakouts in the radial direction was observed within the limit of the uncertainty of our analysis. However, width increases beyond the uncertainty of our determination were highlighted. This contradict the common assumption that breakouts do not widen but only deepen until the borehole reach a new stable state (Zoback et al. 2003);

(iii)  The changes in breakout width occur between datasets collected prior and after reservoir stimulation. However, the most likely effect on breakout width is the thermal equilibration of the wellbore and our data do not evidence stress changes result from reservoir stimulation;

In addition to this analysis, the study of the geometry of borehole failures in both wells leads to propose a characterization of the *in-situ* stress tensor at depths including the orientation and the magnitude of the three principal stresses. This detailed stress state analysis includes the estimation of thermal stresses. A Mohr-Coulomb criterion is retained here to estimate the principal stresses magnitude as it is in our parametrization, the most consistent with a frictional strength limit in the crust. The strength parameterization is however uncertain due to the lack of mechanical testing on the Rittershoffen reservoir rocks. Given the uncertainties, we propose the following careful interpretation of our measurements:

(i)  Our analyses of the breakout geometry variation with depth suggest a change in mean orientation, with a 30° rotation from NNW to NNE highlighted between the bottom and the top of our analysed section. This observation is robust and independent of the strength parameterisation. The rotation does not occur at the sediment-basement interface but is related to a high steeply dipping major fault zone crossing the GRT-1 borehole at a depth of 2368 m (Vidal et al., 2016).





(ii)      Our results suggest also a step in horizontal stress magnitude at the sediment to basement transition that would

be consistent with stiffness contrast between these two lithologies. However, such step is determined by the

choice of the failure criterion and its parameterization which is uncertain at Rittershoffen.

(iii)     $SH$ is generally slightly larger than the vertical principal stresses $Sv$ consistently with a strike-slip to normal

faulting transitional regime. This is consistent with stress characterization at the neighbour site of Soultz-sous-

Forêts (Cornet et al., 2007; Klee & Rummel, 1993; Valley & Evans, 2007b)


The Rittershoffen borehole imaging dataset is unique by the fact that repeating logging allowed to study the temporal
evolution of borehole breakouts and the possible stress changes induced by reservoir stimulation. Our results change the
common view that breakouts mostly deepen but do not widen. Further work is however required to reduce the uncertainties
related to stress magnitude estimates from borehole breakouts and to be able to quantify stress changes induced by reservoir
stimulation.



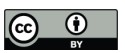

**Availability of data and materials**

Due to the industrial property of the borehole datasets, raw data would remain confidential and would not be shared.

**Competing interests**

The authors declare no competing financial interest.

**Funding**

This work has been published under the framework of the LABEX ANR- 11-LABX-0050-G-EAU- THERMIE-PROFONDE
and benefits from a funding from the state managed by the French National Research Agency (ANR) as part of the
'Investments for the Future' program. It has also been funded by the EU projects DESTRESS (EU H2020 research and
innovation program, grant agreement No 691728).

**Acknowledgments**

We thank ÉS-Géothermie, subsidiary company of Électricité de Strasbourg (ÉS), for support and allowing us the use of
borehole data on wells GRT-1 and GRT-2 of the Rittershoffen ECOGI project. A part of this work was conducted in the
framework of the EGS Alsace project, which was co-founded by ADEME.
We would like to thank the Swiss Competence Center for Energy Research–Supply of Electricity (SCCER-SoE) for support
of the study. The present work has been done under the framework of the LABEX ANR-11-LABX-0050-G-EAU-
THERMIE-PROFONDE and benefits from a state funding managed by the French National Research Agency (ANR) as part
of the "Investments for the Future" program.




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





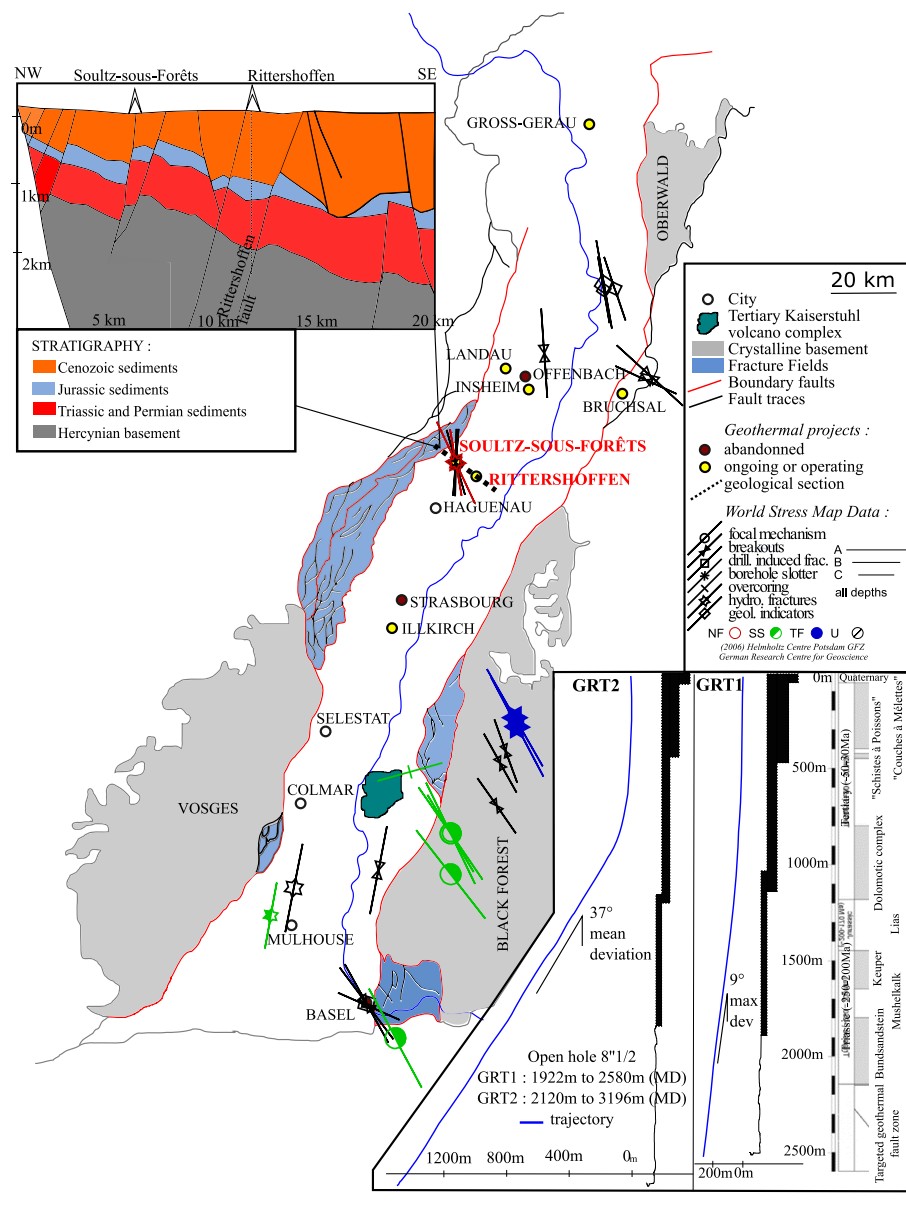



**Figure 1: Geological and structural map of the main of the Upper Rhine Graben with the location of the Rittershoffen and Soultz-sous-Forêts sites. The map shows also the location and status of other neighbouring deep geothermal projects. Stress data from World stress map database (Heidbach et al., 2010) are included. Upper left insert shows a geological section highlighting the main units crossed by the wells in Rittershoffen and Soultz-sous-Forêts (Aichholzer et al., 2016; Baujard et al., 2017). Lower right insert is a sketch of wells GRT-1 and GRT-2 drilled in Rittershoffen, including their geometry, depths and crossed lithology (after Baujard et al. (2015, 2017)).**





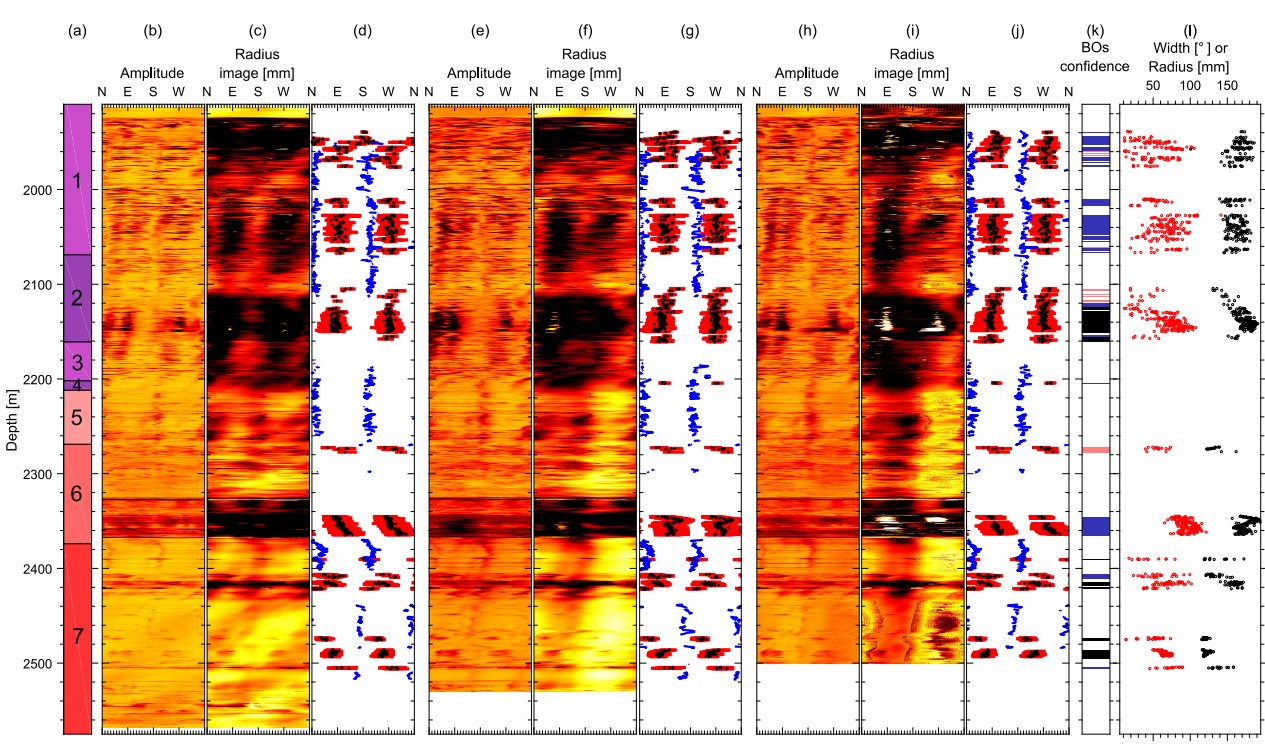



**Figure 2: Synthesis of the data used in this analysis of the borehole GRT-1. (a) simplified lithologic column. (1) stands for "couches de Rehberg", (2) for "Couches de Trifels", (3) for Annweiler sandstone, (4) for Permian layers older than Annweiler sandstone, (5) for rubefied granite, (6) for hydrothermally altered granite and (7) for low altered granite. The UBI images are presented, as well as the data picked from the visual analysis of the double transit time image for the dataset of 2012 (panel b. - c. - d.), 2013 (e. - f. – g.), and 2015 (h. - i. - j.) collected in GRT-1. The radius of the borehole computed from the double transit time image is displayed in panels b. - e. and h. In panels d. - g. and j., blue dots represent the azimuth of the Drilling Induced Tension Fractures (DITFs), black dots represent the azimuth of the maximal radial depth of the breakouts and red bars represent the extension between the edges of the breakouts. Panel l. summarizes the width (black dots, in °) and the enlargement radius (red dots, in mm) measured in the 2012, 2013 and 2015 images and panel k. informs about the breakouts (BOs) confidence level applied to these results.**





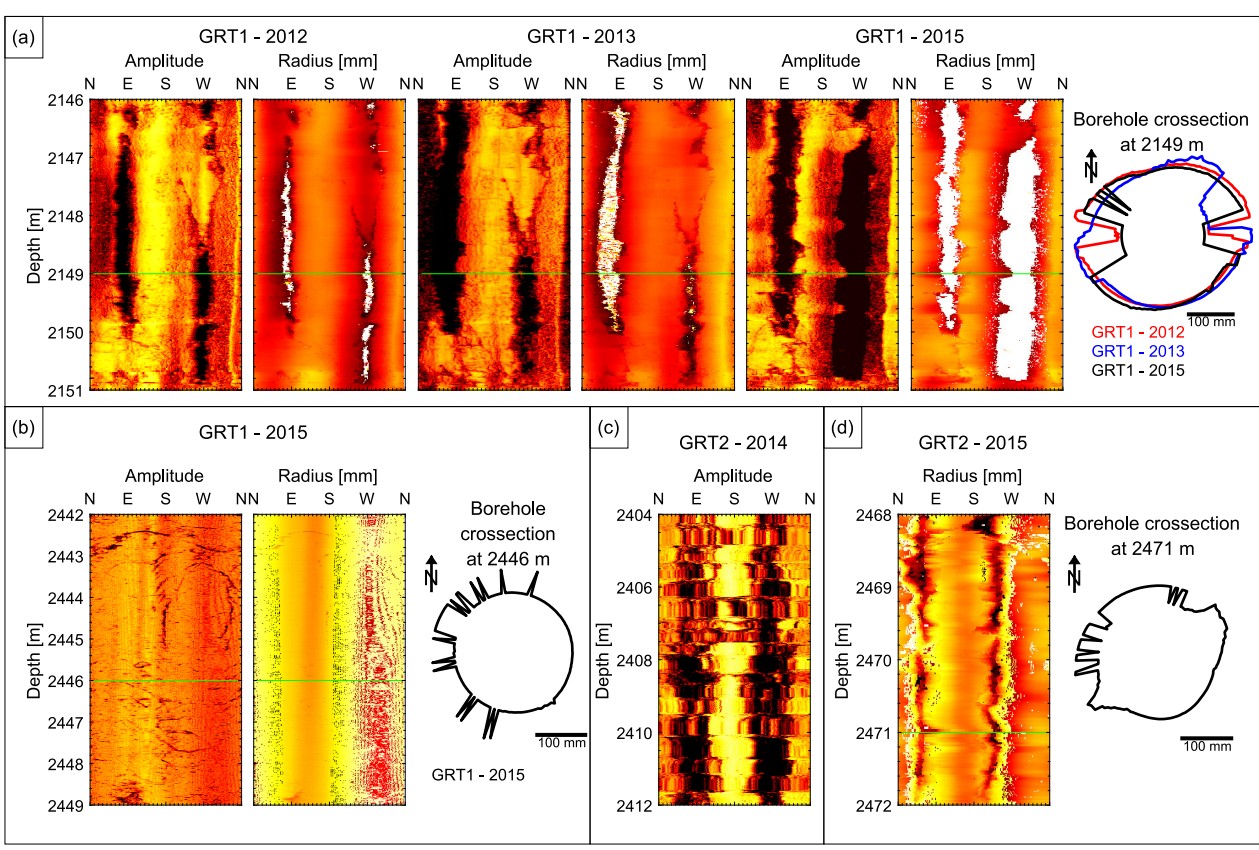





**Figure 3: Example of image artefact observed on the GRT-1 and GRT-2 data set. a) Comparison of data from 2012, 2013 and 2015 collected in GRT-1 presenting a signal loss artefact in sandstones. b) Wood grain artefact visible on the 2015 GRT-1 image, both on the amplitude and radius image in granite. This leads to noisy borehole section. c) Stick-slip artefact present along the entire GRT-2-2014 image. d) Erroneous radius record observable on the GRT-2-2015 image in granite, possibly related to tool decentralization.**





**a)** Search Box

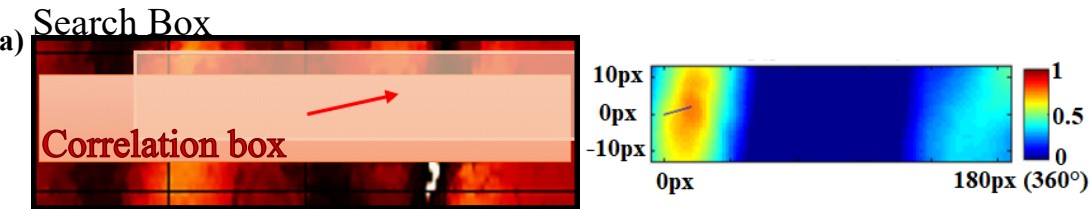

**b)**

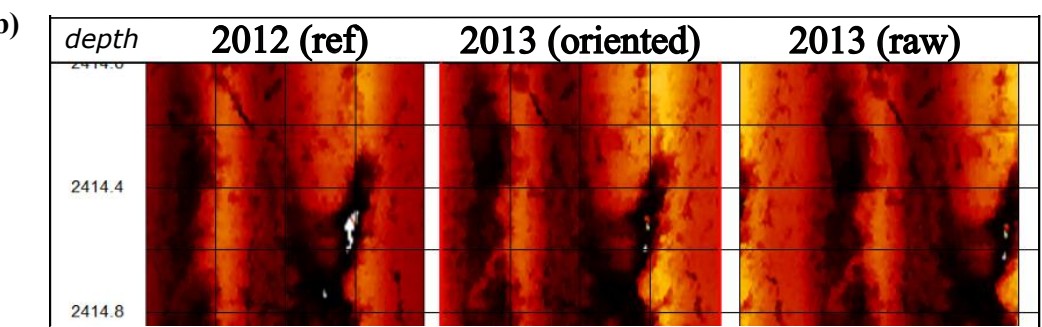





**Figure 4: a) Sketch presenting the process used to orientate the images of GRT-1. A correlation box is defined in the double transit time image of reference (acquired in 2012) and is progressively shifted in the image to compare with (red windows) within the limits of the search box (black window). We compute the correlation between the correlation box in its initial position in the image of reference and the shifted correlation box in the image to compare with for each position (right insert). The displacement maximizing the correlation factor enables, at a given depth, to rotate and adapt the image of 2013 and 2015 according to the image of 2012. b) example of original and reoriented time transit images of 2013, at a depth of 2414m (TVD) in GRT-1.**





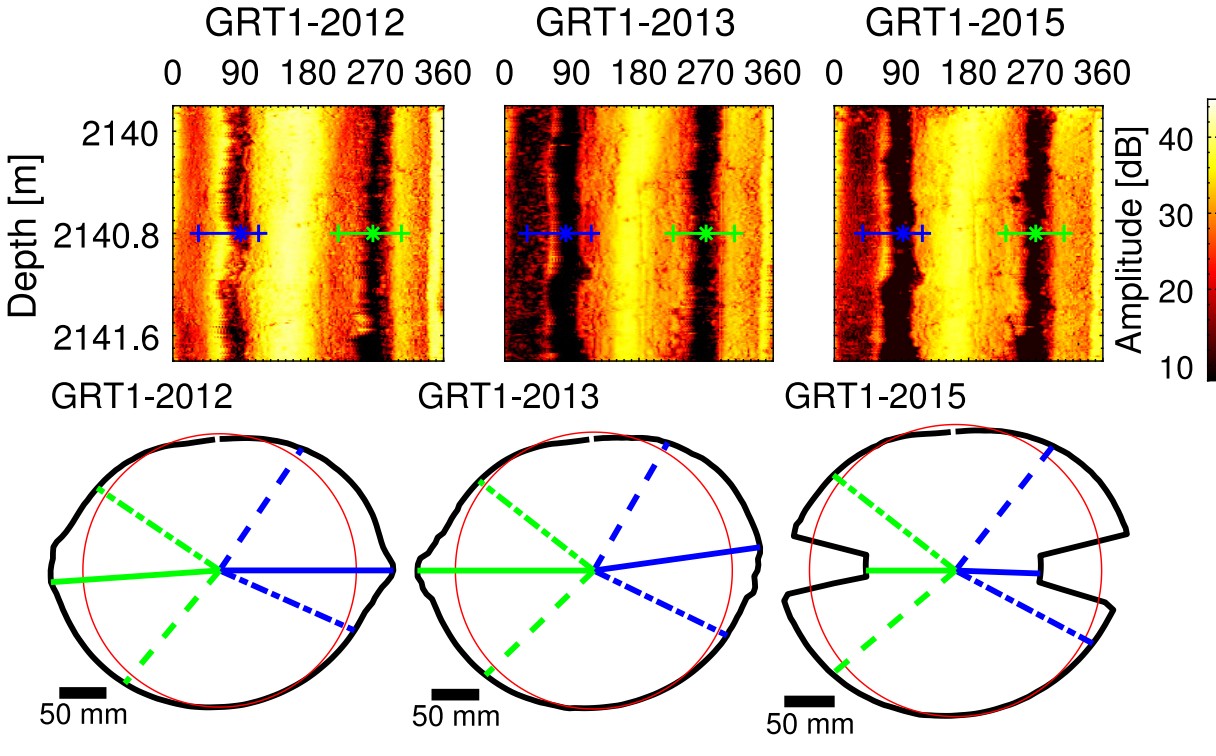





**Figure 5: Example of breakout geometry determination in sandstones. Upper figures: amplitude images for GRT-1 at 2140.8 m for the logs from 2012, 2013 and 2015. Lower figures: wellbore section at 2140.8 m computed from the transit time images from the 2012, 2013 and 2015 logs respectively. The breakout extent is determined on the wellbore section. The blue and green dashed lines represent the extent of the breakout when the plain lines represent the azimuth of maximum radial extension of the breakout.**





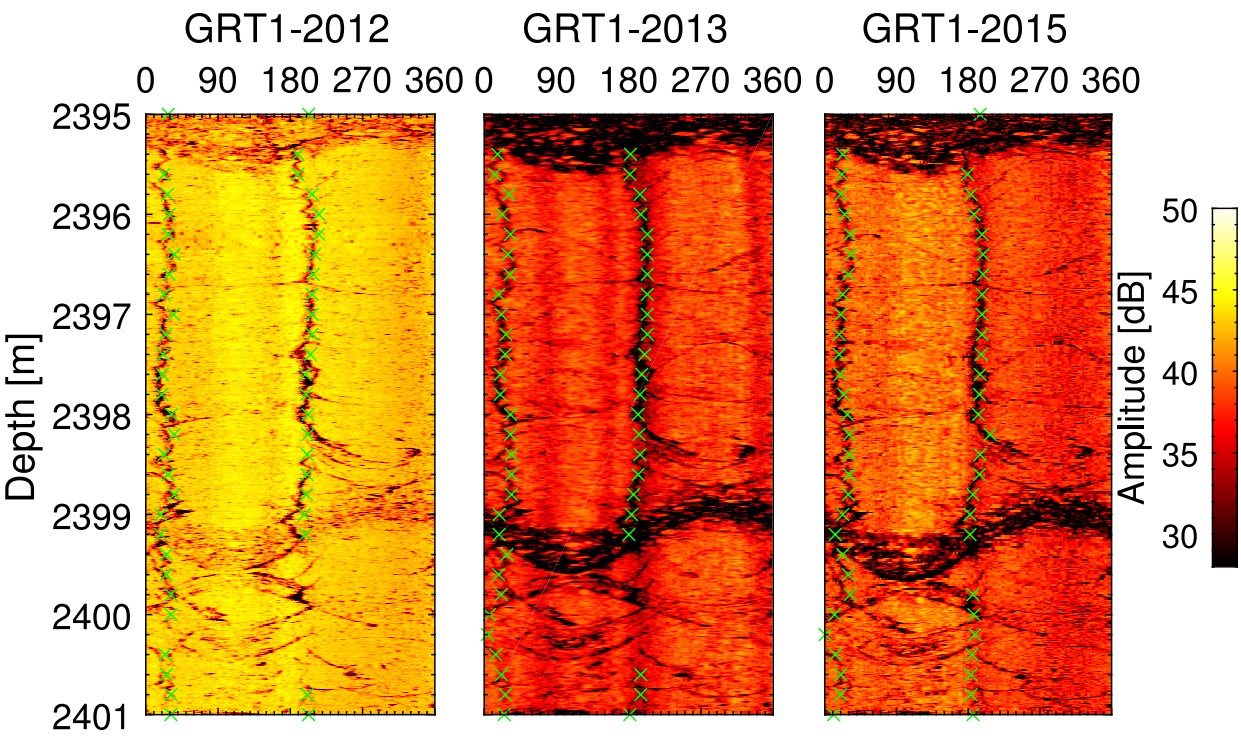





**Figure 6: Examples of Drilling Induced Tension Fractures (DITFs), observed in the granitic section of GRT-1 in the amplitude images acquired in 2012, 2013 and 2015. The azimuth of the DITFs is measured every 20 cm (green crosses).**







**Figure 7: Examples of breakout shape evolution between the three successive images collected in GRT-1 in sandstones. Upper figures show the amplitude images and the radius computed from the time transit images for a section of GRT-1 from 2124 to 2128m (TVD) in 2012, 2013 and 2015. Lower figures show the mean section computed at 2125.6 and 2126.2m (TVD) from the time transit images averaged over 60cm intervals. The sections are represented along with an 8.5 inch radius circle representing the unaltered open hole section. The sections from the image of 2012, 2013 and 2015 are superposed in the right panel.**



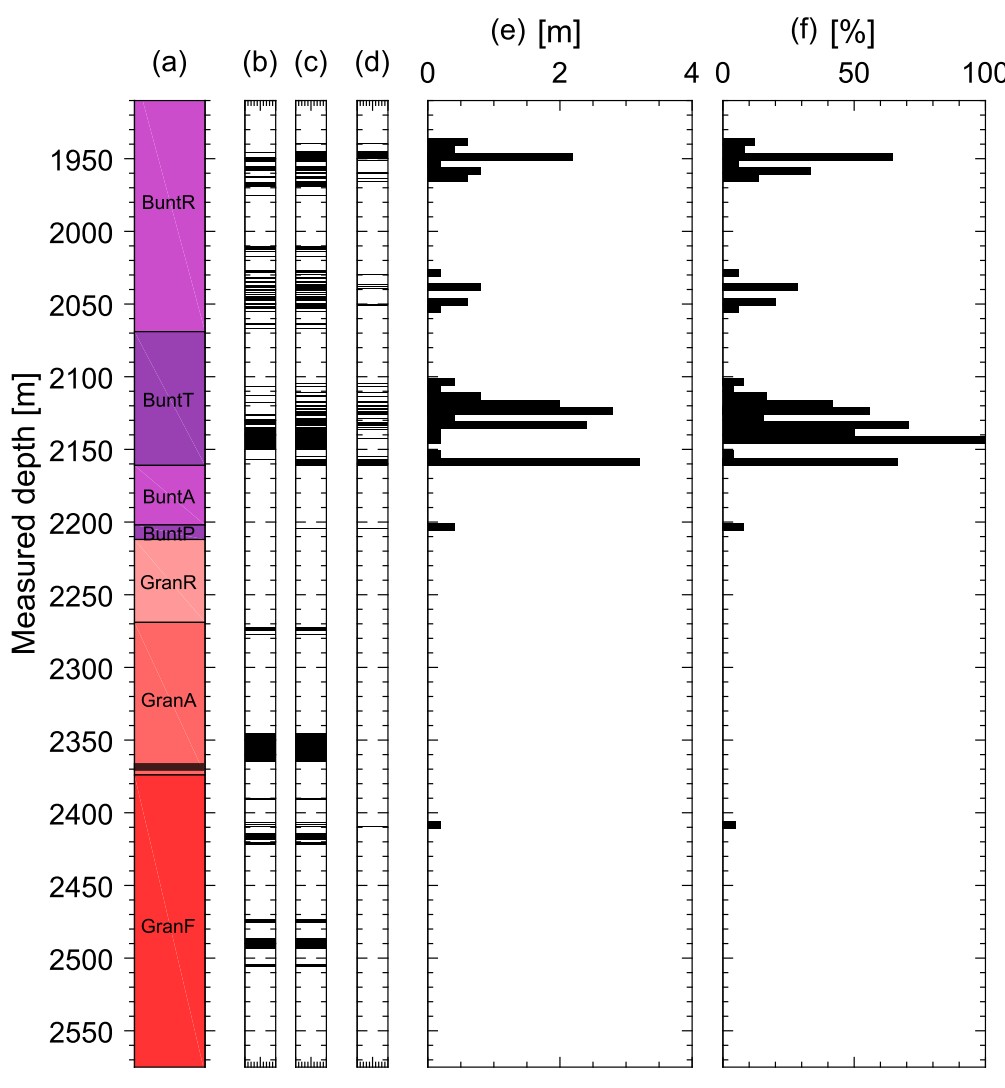



**Figure 8: Development of breakouts along GRT-1 borehole between 2012 and 2013. a) Simplified lithologies along GRT-1 borehole with depth (MD): BuntR stands for "couches de Rehberg", BuntT for "Couches de Trifels", BuntA for Annweiler sandstone, BuntP for Permian layers older than Annweiler sandstone, GranR for rubefied granites, GranA for hydrothermally altered granite and GranF for low altered granite. The major fault zone crossing GRT-1 at 2368m is represented as a black band. b) Breakouts positions in GRT-1 in 2012. c) Breakouts positions in GRT-1 in 2013. d) Intervals where breakouts are present in 2013 but not in 2012. e) Breakout length increase in [m] along the borehole between 2012 and 2013 in 5 m bins. f) fraction in [%] of wellbore length that was free of breakout in 2012 that is presenting breakout on the 2013 image, computed in 5 m bins.**





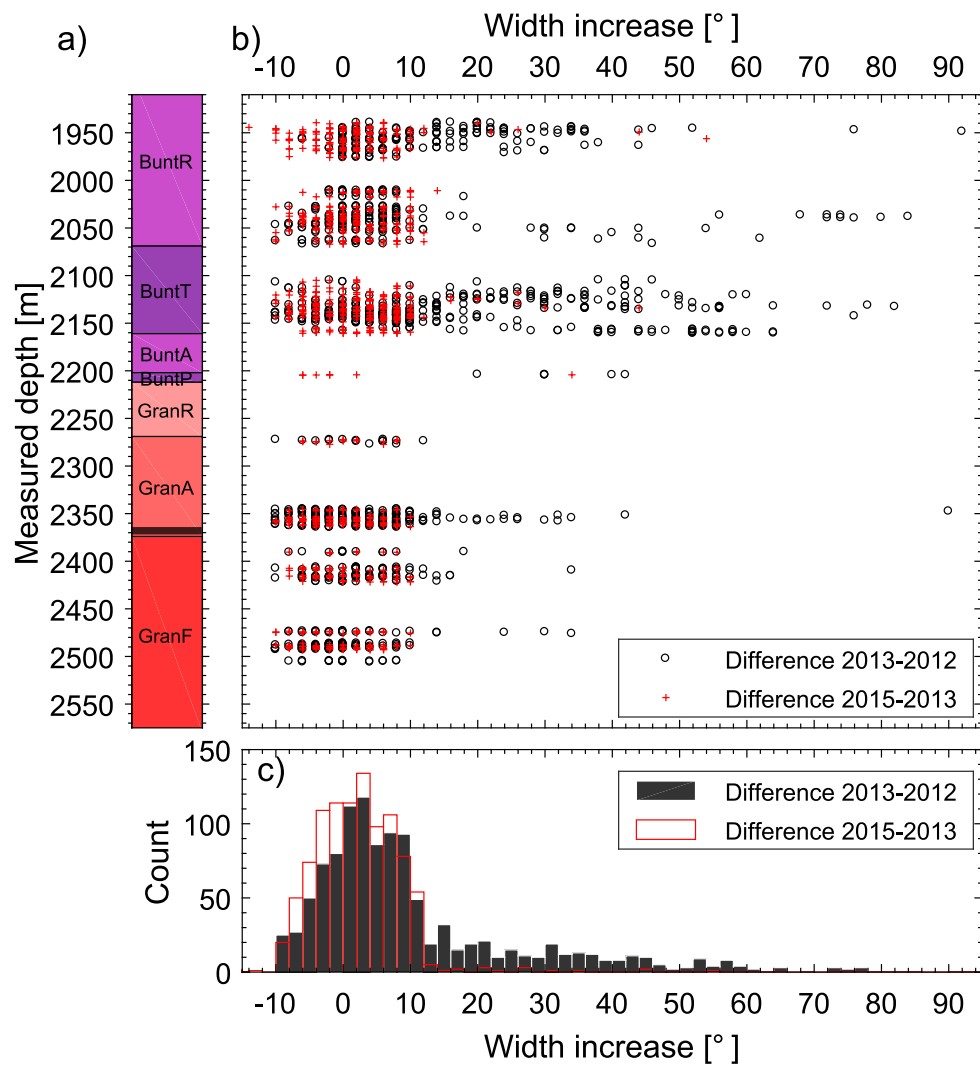




**Figure 9: Evolution of breakout width in GRT-1 borehole with depth (MD). a) Simplified lithologies along GRT-1 borehole (see Fig. 8 for the legend). b) Width increase between the 2012-13 time interval (black circles) and the 2013-15 time interval (red crosses) presented as a function of the vertical depth. c) histograms in 2° classes of breakout width changes for the 2012-13 interval (black) and the 2013-15 interval (red).**





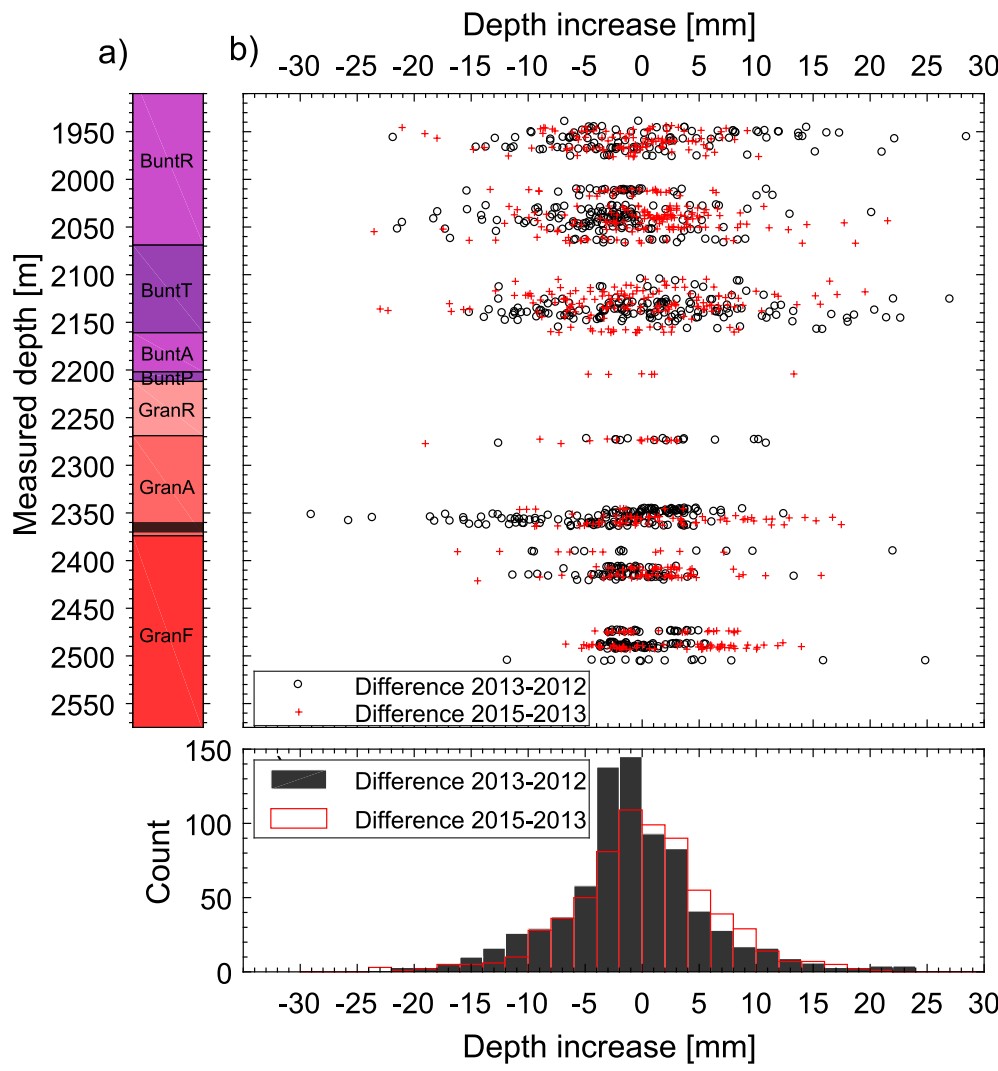





**Figure 10: Evolution of the depth of the breakouts in the GRT-1 borehole with depth (MD). a) Simplified lithologies along GRT-1 borehole (see Fig. 8 for the legend). b) Increase of the maximum radial extension between the 2012-13 time interval (black circles) and 2013-15 time interval (red crosses) presented in function of depth. c) histograms in 2 mm classes of breakout with changes for the 2012-13 interval (black) and 2013-15 interval (red).**





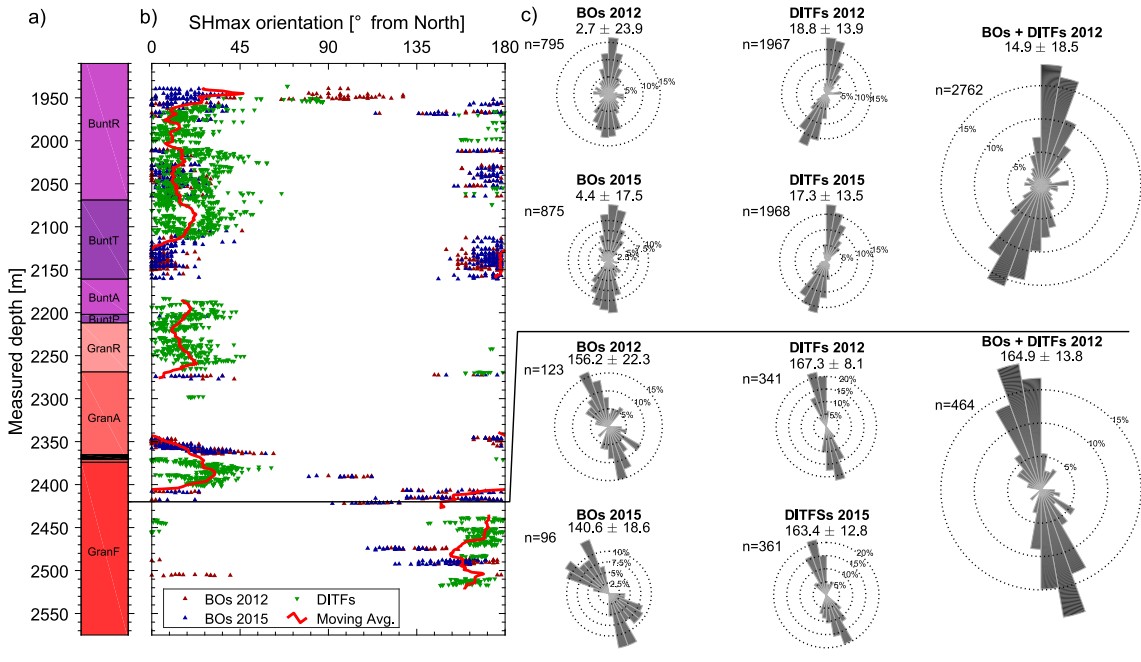



**Figure 11: Evolution in orientation of the maximum principal stress with measured depth in GRT-1, in 2012 and 2015. a) Simplified lithologies along GRT-1 borehole (see Fig. 8 for the legend). b) Orientation of *SH* from the azimuth of maximum radial extension of the breakouts (BOs) from the dataset of 2012 (in blue) and of 2015 (in red) acquired in GRT-1. In green, orientation of *SH* from the azimuth of Drilling Induced Tensile Fractures (DITFs). The red line is a moving average of the orientation data. c) From the datasets displayed in panel b), orientation in rose diagrams.**





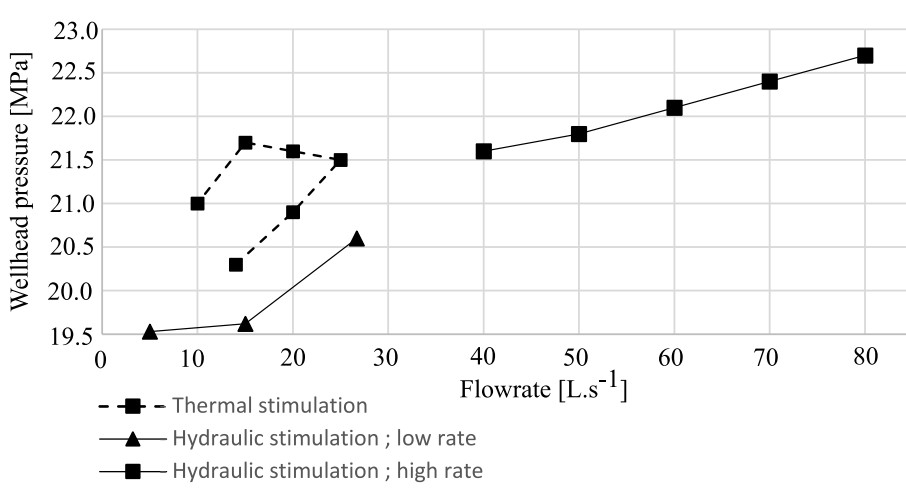





**Figure 12: Minimal horizontal stress *Sh* [MPa] with depth at the Soultz-sous-Forêts site from three studies obtained from high-volume injections in GPK-1, GPK-2, GPK-3 and EPS-1 wells. The lower bound for the minimal horizontal stress *Sh* obtained from the analysis of the hydraulic stimulation of the well GRT-1 is represented for comparison (black circle).**





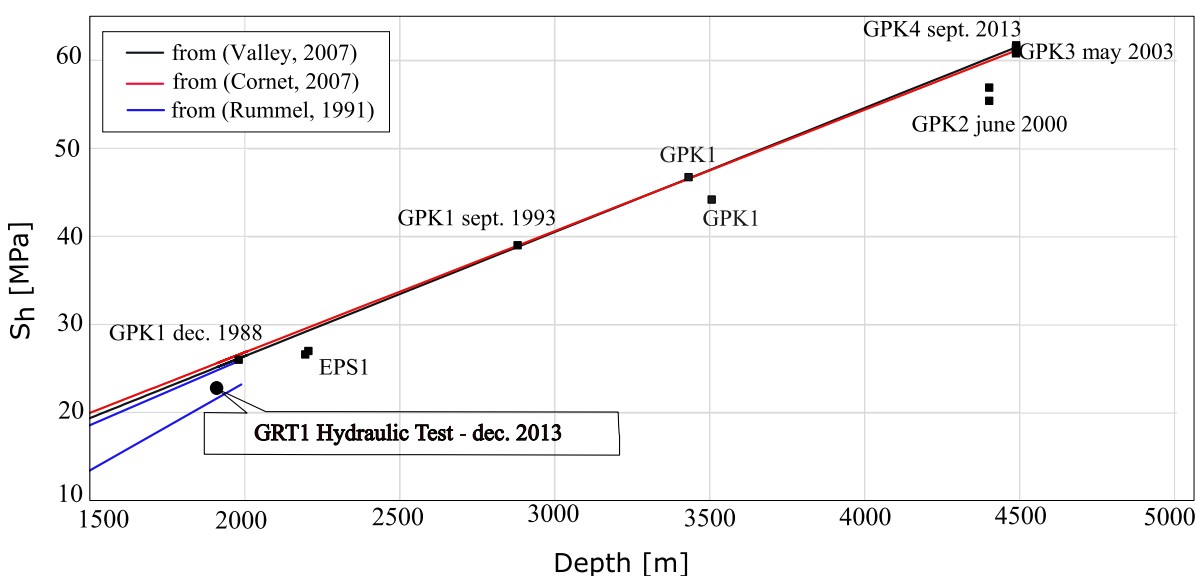




**Figure 13: Stabilized wellhead pressure [MPa] for each flowrate step measured during the stimulation operations conducted in GRT-1 in 2013, targeting a depth of 1913 m (TVD), as a function of flow rate [L.s$^{-1}$] (after Baujard et al., 2017).**



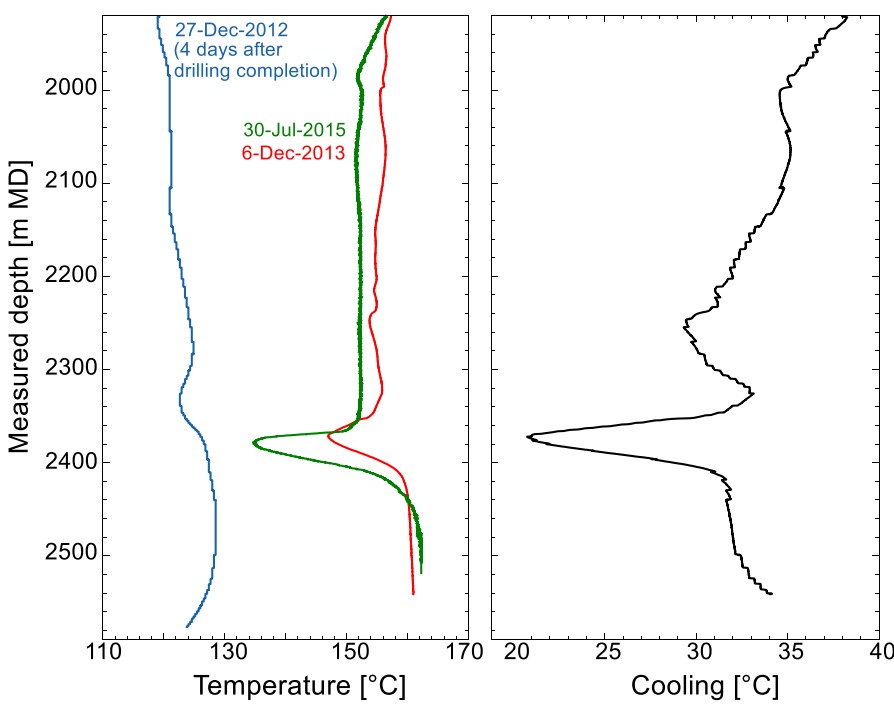



**Figure 14: Left panel: variation of temperature [°C] with depth (MD), estimated from the temperature log acquired in 2015 in GRT-1 (green curve), plotted along with the temperature log acquired in 2013 (red curve). The temperature log acquired four days after drilling completion (blue curve) enables to estimate the temperature at the borehole wall during drilling. Right panel: estimation of the difference in temperature between the wellbore temperature and the borehole wall temperature after completion Δt used in the evaluation of the thermal stress components.**





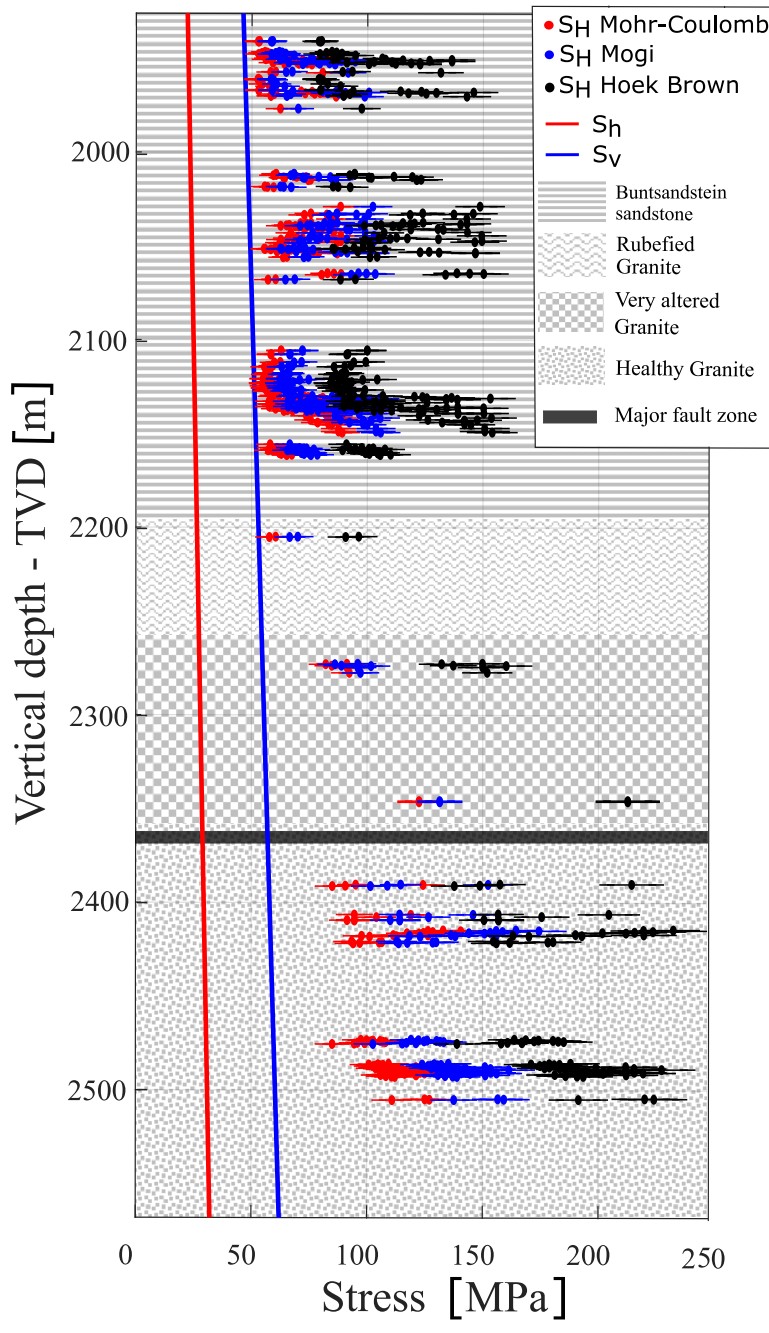





**Figure 15:** *in-situ* **stress state components** *Sh***,** *SV* **and** *SH* **[MPa]. Maximum horizontal stresses** *SH* **are inverted with three distinctive failure criteria for the images of 2013 of GRT-1 well. Error bars are calculated considering the error on the measurement of the breakout width, on the estimates of the elastic parameters and on the** *Sh* **and** *SV* **trends with depth. The background pattern represents the four major lithological units retained in the model and the major fault zone crossed by GRT-1.**



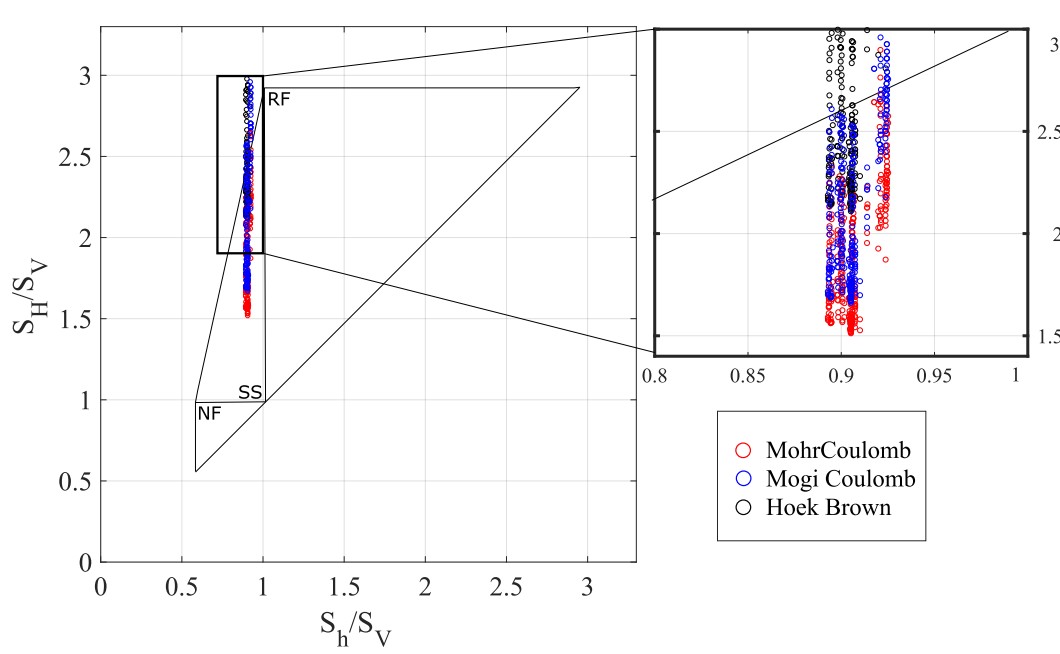





**Figure 16: Normalized stress polygon defining stress states (*SH/SV, Sh/SV*) at a depth of 2500m in GRT-1, according to a Coulomb law with a coefficient of friction *μ*=1. The borders of the polygon correspond to an active fault situation. According to Anderson's faulting theory, *RF* – reverse faulting – *SS* – strike slip regime – and *NF* – normal faulting – refer to the Anderson's faulting regimes. It is plotted along with the stresses (*SH/SV – Sh/SV*) calculated from the image of the GRT-1 of 2013, for three different failure criteria (circles in color).**





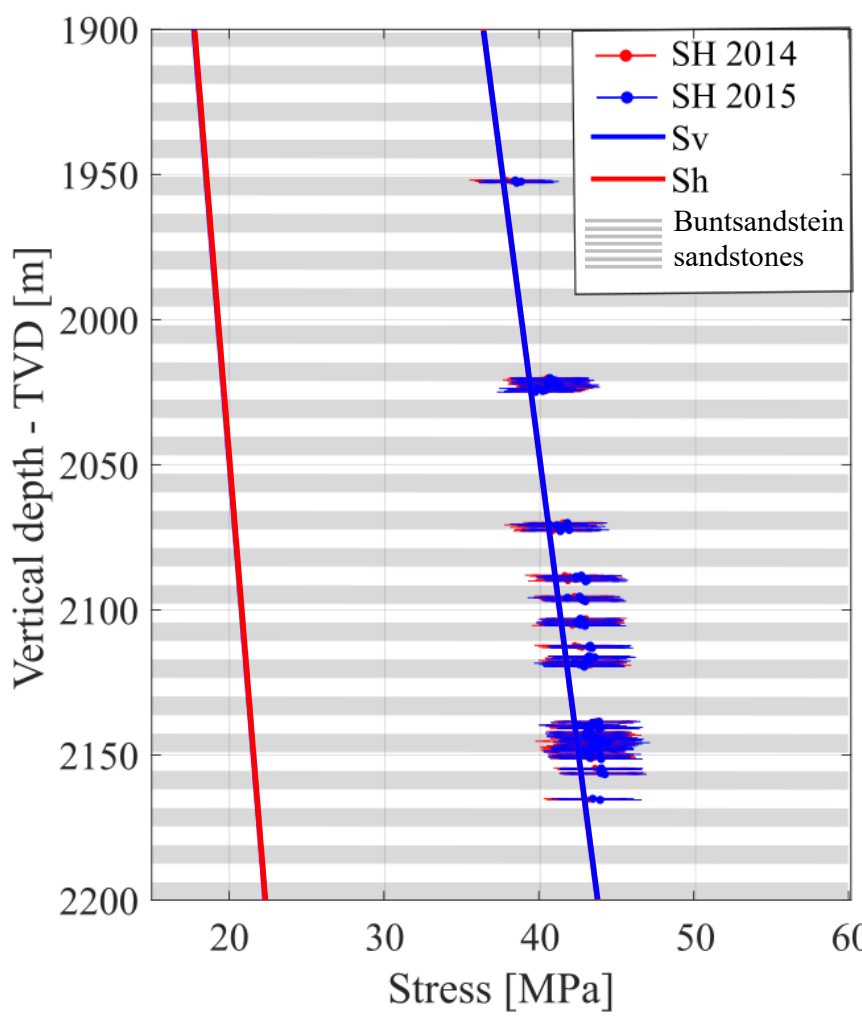



**Figure 17: in-situ stress components *Sh*, *SV* and *SH* [MPa] in the deviated well GRT-2. *SH* stresses are inverted using a Mohr Coulomb failure criterion and represented as a function of the vertical depth for the images acquired in 2014 and 2015. Error bars are calculated considering the errors on the measurements of the breakout widths, on the elastic parameters and on the *Sh* and *Sv* trends. The background pattern represents the major lithological unit crossed by the well.**



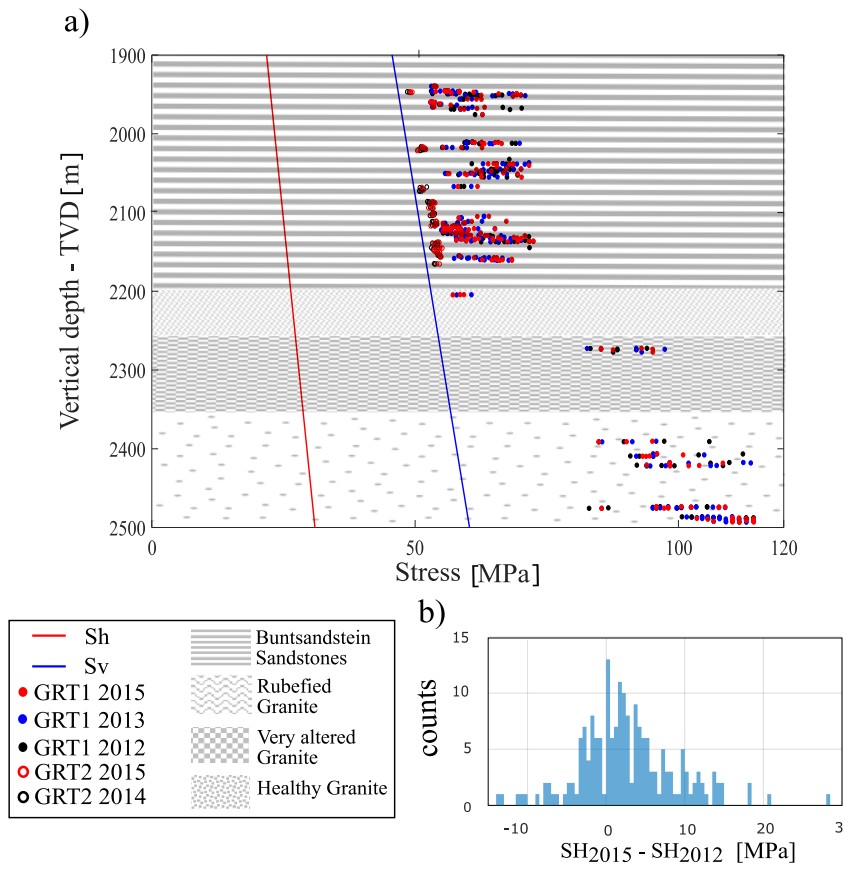





**Figure 18: Panel a. shows the *SH* stresses [MPa] inverted with a Mohr-Coulomb criterion from the images acquired in 2012 – 2013 and 2015 (plain circles) in GRT-1 and in 2014 and 2015 (empty circles) in GRT-2, as a function of vertical depth. The background shows the lithological units retained in the model. Panel b. shows the difference in *SH* stresses between 2015 and 2012 in GRT-1 in a histogram with 1 MPa bins.**



**Table 1: Data acquired in GRT-1 and GRT-2 and specificities of UBI acquisition programs.**

| Well | Acquisition Date | Stimulation | Logging depth range [m; MD] | Transducer diameter [inch] |
|---|---|---|---|---|
| GRT1 | 30-Dec-2012 | 4 days after drilling completion | 1913.00 - 2568.00 | 4.97 |
| | 9-Dec-2013 | 1 year after drilling completion 5 months after THC stimulation | 1912.00 - 2531.16 | 2.92 |
| | 30-Jul-2015 | 2.5 years after drilling completion 2 years after THC stimulation. | 1910.96 - 2499.9 | 4.97 |
| GRT2 | 23-Jul-2014 | Four days after drilling completion | 2118.00 - 2531.22 | 4.97 |
| | 29-Jul-2015 | 1 year after drilling completion | 2111.00 - 2869.23 | 4.97 |

**Table 2: Elastic (Poisson ratio) and strength parameters (used in the Mohr-Coulomb, Mogi-Coulomb and Hoek Brown failure criteria) for the four geological units retained in the model, for both GRT-1 and GRT-2 wells, as a function of measured depth (MD). Elastic and strength parameters for granites are based on a data compilation of tests conducted on samples from Soultz-sous-Forêts. For the Buntsandstein sandstones, we use usual strength parameters based on Hoek & Brown (1997).**

| Depth (MD) [m] GRT1 | Depth (MD) [m] GRT2 | Geology | | Elastic and strength Parameters | | | | | |
|---|---|---|---|---|---|---|---|---|---|
| | | Stratigraphy | Lithology | $\nu$ | Cohesion $C$ [MPa] | Internal Friction $\theta$ | UCS [MPa] | (a, b) Mogi Coulomb | Hoek Brown $mi$ |
| 1799-2212 | 2022-2479 | Bunt-sandstein | Sandstones (argilic) | 0.22 | 24 ±5 | 35° | 92±14 | (18 ±3, 0.54) | 19 |
| 2212-2269 | 2479-2629 | Granitic Basement | Ruberfied Granite | 0.26 | 23 ±5 | 40° | 100 ±15 | (13 ±3, 0.68) | 20 |
| 2269-2374 | 2629-2881 | | Hydrothermally altered Granite | 0.26 | 29 ±5 | 40° | 125 ±17 | (17 ±3.5, 0.68) | 23 |
| 2374-2580 | 2881-3196 | | Low altered Granite | 0.26 | 32 ±5 | 45° | 155 ±20 | (21 ±3.5, 0.68) | 27 |



**Table 3: Mean density retained for each lithological layer and vertical depth (TVD) in each well.**

| Description | Depth in GRT1 [m] | Depth in GRT2 [m] | Volumetric mass [kg.m⁻³] |
|---|---|---|---|
| Tertiary | 0 | 0 | 2350 |
| | 1172 | 1166.5 | |
| Jurassic | 1172 | 1166.5 | 2440 |
| | 1447 | 1431.5 | |
| Keuper | 1447 | 1431.5 | 2700 |
| | 1653 | 1637 | |
| Muschelkalk | 1653 | 1637 | 2750 |
| | 1798 | 1793.5 | |
| Top Buntsandstein | 1798 | 1793.5 | 2610 |
| | 1855 | 1850 | |
| Mean Buntsandstein | 1855 | 1850 | 2520 |
| | 2147 | 2109 | |
| Bottom Buntsandstein | 2147 | 2109 | 2540 |
| | 2198 | 2167 | |
| Granitic basement | 2198 | 2167 | 2570 |
| | 2568 | 2707.5 | |