# Peer review of "Stress Characterization and Temporal Evolution of Borehole Failure at the Rittershoffen Geothermal Project"

_Solid Earth, 2019_

## Referee Comment (RC1) · Francois Cornet (Referee) · 18 Apr 2019

This paper addresses the important issue of evaluating a regional stress field from images of two different failure processes (borehole breakouts and so called drilling Induced fractures) observed in deep boreholes with different orientations, as well as from results from various water injection tests. The methodology is applied at the Rittershoffen site, located 6km east from the Soultz site, where the stress field is quite well known. This is an important contribution for the understanding of stress field in deep rock masses and the quality of images as well as that of their analysis justify completely its publication. But before publication of the paper, some errors must be corrected and

the discussion of results must be revised. Here after my main comments. 1. The GRT-2 borehole is inclined 37° to the vertical so that the axial and tangential stress components at the borehole wall are not principal stresses. Authors must write down the equations they are considering, including the role of pore pressure, and that of thermal stresses. Indeed the principal directions, at the wellbore wall, of stresses resulting from the far field stresses are not the same as those of the thermal stresses resulting from the cooling of the rock. This issue is completely ignored and the paper cannot be published before this is properly dealt with. I encourage authors to look at paper by Wileveau et al. that provides good illustrations of en echelon breakouts observed in inclined wells. (Wileveau Y , F.H. Cornet, J. Desroches and P. Blumling, 2007 ; Complete in situ stress determination in an argillite sedimentary formation; Physics and Chemistry of the Earth (vol. 32, pp 866-878) 2. For their analysis of the width of borehole breakouts, authors refer to three different failure criteria, including the Hoek and Brown criterion. For the parameters to be considered in these criteria, they refer to laboratory work quoted by Rummel, 1991 and by Valley and Evans, 2006. They should also look at the publication by Villeneuve et al. (Villeneuve M.C., M.J. Heap, A.R.L. Kushnir, T. Qin, P. Baud, G. Zhou, and T. Xu, 2018; Estimating in situ rock mass strength and elastic modulus of granite from the Soultz-sous-forêts geothermal reservoir (France); Geothermal Energy, 6(11), https://doi.org/10.1186/s40517-018-0096-1), which address precisely this issue. 3. In their table 3 the density value for the granite is said to be 2570 kg/m3, yet in equation (6) the vertical stress is assumed to be equal to 0.024 z-0.83. These differences should be discussed. In addition, given the vertical stress magnitude is taken into consideration in the three dimensional failure criteria, authors should show how they determine uncertainties on the vertical stress component evaluation. 4. Similarly, equations used for the evaluation of the minimum principal stress magnitude is not described and this should be corrected. Evaluation of associated uncertainty should be discussed. 5. Table 2 indicates values for the Poisson's ratio but no reference is made to Young's moduli nor to thermal expansion coefficients used in equation 8. How are the various parameters measured? How valid

are those measurements for in-situ properties ? This should be better discussed. 6. In equation (2) the stress component $\tau$oct implies the three principal stress components. This should also apply to the mean stress, as opposed to equation written on line 179. 7. In their discussion of results, authors argue that some of the results obtained for the magnitude of the maximum principal stress magnitude do not satisfy the Coulomb stability condition for the rock mass. Interestingly, Cornet (2016) has argued that the large scale fluid injections conducted at Soultz have generated large scale failure zones that are changing in orientation with depth, a feature consistent with the Hoek and Brown criterion but not with a Coulomb criterion. This issue should be discussed more carefully (Cornet, F.H., 2016. Seismic and aseismic motions generated by fluid injections; Geomech. Ener. Env., 5, pp 42-54).

caption of figure 12 has been exchanged with that of fig 13

Because of these many issues, I recommend publication of this paper only after they have been answered, with particular attention to the issue raised on principal stress directions close to inclined boreholes.

―――――――――――――――――――

---

## Author Comment (AC1) · 29 Apr 2019

We thank François Cornet for his comments and review. We appreciate his recognition of the importance of our contribution. Please find below a point by point response to the comments. A pdf file including in black the comments and in blue, our response, is provided as supplement.

Sincerely, on behalf of the authors Jérôme AZZOLA

This paper addresses the important issue of evaluating a regional stress field from images of two different failure processes (borehole breakouts and so-called drilling

[Figure]

Induced fractures) observed in deep boreholes with different orientations, as well as from results from various water injection tests. The methodology is applied at the Rittershoffen site, located 6km east from the Soultz site, where the stress field is quite well known. This is an important contribution for the understanding of stress field in deep rock masses and the quality of images as well as that of their analysis justify completely its publication.

1. The GRT-2 borehole is inclined 37° to the vertical so that the axial and tangential stress components at the borehole wall are not principal stresses. Authors must write down the equations they are considering, including the role of pore pressure, and that of thermal stresses. Indeed, the principal directions, at the wellbore wall, of stresses resulting from the far field stresses are not the same as those of the thermal stresses resulting from the cooling of the rock. This issue is completely ignored, and the paper cannot be published before this is properly dealt with. I encourage authors to look at paper by Wileveau et al. that provides good illustrations of en echelon breakouts observed in inclined wells. (Wileveau Y, F.H. Cornet, J. Desroches and P. Blumling, 2007; Complete in situ stress determination in an argillite sedimentary formation; Physics and Chemistry of the Earth (vol. 32, pp 866-878)

We acknowledge that the GRT-2 is strongly inclined, with a mean deviation of 37° measured in the section of interest. The equations describing the stress concentration at the borehole wall of a vertical borehole, used in particular for the well GRT-1, are no longer applicable in this case. For the deviated well GRT-2, we used a 3D solution taking into account the geometry of the borehole. The equations in which are involved the geometrical parameters, the far field stresses and the fluid pressure are well documented in the literature and we used the summary proposed in the review from Schmitt et al. (2012) who proposes a complete development of the equations, in the general case. We will include the computation steps leading to the expression of the effective principal stresses at the borehole wall of the deviated well in the revised version of the manuscript and we will cite the work of Wileveau et al., as an additional reference to

this approach.

2. For their analysis of the width of borehole breakouts, authors refer to three different failure criteria, including the Hoek and Brown criterion. For the parameters to be considered in these criteria, they refer to laboratory work quoted by Rummel, 1991 and by Valley and Evans, 2006. They should also look at the publication by Villeneuve et al. (Villeneuve M.C., M.J. Heap, A.R.L. Kushnir, T. Qin, P. Baud, G. Zhou, and T. Xu, 2018; Estimating in situ rock mass strength and elastic modulus of granite from the Soultz-sous-Forêts geothermal reservoir (France); Geothermal Energy, 6(11), https://doi.org/10.1186/s40517-018-0096-1), which address precisely this issue.

We used all available published data to parametrize our failure criteria, including data provided in Villeneuve et al., 2018, but also from Heap et al. (2019) (cited on lines 90 and 511 of our original manuscript). To clarify this, we will add the relevant references in section 5 of our revised manuscript.

3. In their table 3 the density value for the granite is said to be 2570 kg/m3, yet in equation (6) the vertical stress is assumed to be equal to 0.024 z-0.83. These differences should be discussed. In addition, given the vertical stress magnitude is taken into consideration in the three-dimensional failure criteria, authors should show how they determine uncertainties on the vertical stress component evaluation.

The magnitude of the vertical stress Sv is obtained from the weight of the overburden. We apologize for the typo in Eq. (6), which should read 0.248 z – 0.83. This misleading rounding will be corrected in the revised manuscript, which leads to a trend in line with density value 2570 kg/m3 chosen for the granitic layer. Given the fact that the vertical stress is obtained by integrating the density profile from surface to reservoir depth, the uncertainty on density add up and thus the uncertainty on the vertical stress estimation increase with depth. Considering an uncertainty of 50 kg/m3 on the densities leads to a 2.5 MPa uncertainty on the vertical stress at reservoir depth. This uncertainty is not significant compared to other uncertainties involved in the analysis as for example

those related to the mechanical parameters chosen in the inversion of the maximum horizontal stress.

4. Similarly, equations used for the evaluation of the minimum principal stress magnitude is not described and this should be corrected. Evaluation of associated uncertainty should be discussed.

We follow approaches largely used in the literature (e.g. Cornet et al., 2017) and estimate the minimum horizontal stress Sh from pressure limiting behavior during hydraulic injections. Since we did not have enough information related to the ECOGI project to compute a complete Sh stress profile, we use measurements carried out at the nearby Soultz-sous-Forêts project. The trend is evaluated by Cornet et al. (2007). This publication does not propose an uncertainty measurement for the minimum horizontal stress Sh. To complete our analysis, we analyzed the wellhead pressure measured during the hydraulic stimulation of GRT-1 and derived an estimate of the Sh at depth from the pressure reached at a maximum flow rate. However, as the pressure shows a gradual but not definitive stabilization for these maximum flow rates, our measurement is discussed as a lower bound of the minimum horizontal stress Sh at depth. We show that our measurement is still consistent with the trend retained from Soultz.

5. Table 2 indicates values for the Poisson's ratio, but no reference is made to Young's moduli nor to thermal expansion coefficients used in equation

We apologize for failing to include these values in the manuscript. They will appear in the revised version. The thermal expansion is chosen to be constant for the different layers of our model, $\alpha$ = 15 x 10-6 K-1, and the Young's modulus will appear in the Table 2.

6. In equation (2) the stress component $\tau$oct implies the three principal stress components. This should also apply to the mean stress, as opposed to equation written on line 179.

We would like to refer the reviewer to the original derivation of the equations proposed by Zimmerman & Al-Ajmi (2006) in which they refer to an "effective mean stress", $\sigma_{m,2}=\frac{\sigma_1+\sigma_3}{2}$ ,for the Mogi-Coulomb criteria. This is not strictly speaking the mean stress, which would also include a contribution of the intermediate stress. We will clarify the terminology and nomenclature in the revised version of the manuscript.

7. In their discussion of results, authors argue that some of the results obtained for the magnitude of the maximum principal stress magnitude do not satisfy the Coulomb stability condition for the rock mass. Interestingly, Cornet (2016) has argued that the large-scale fluid injections conducted at Soultz have generated large scale failure zones that are changing in orientation with depth, a feature consistent with the Hoek and Brown criterion but not with a Coulomb criterion. This issue should be discussed more carefully (Cornet, F.H., 2016. Seismic and aseismic motions generated by fluid injections; Geomech. Ener. Env., 5, pp 42-54). caption of figure 12 has been exchanged with that of fig 13 Because of these many issues, I recommend publication of this paper only after they have been answered, with particular attention to the issue raised on principal stress directions close to inclined boreholes.

We consulted the proposed paper, but it does not refer to these criteria. We would appreciate it if the reviewer could clarify his comment before we review the paper and explain the link between the large-scale failure zones created by the fluid injections and the consistency with the Hoek Brown criterion rather than with the Coulomb criterion. We should also point out that the injection in Rittershoffen were not as "massive" as in Soultz, i.e. the injected volumes were much smaller and thus the transposition of the knowledge from Soultz in this regard may not be directly applicable.

8. How are the various parameters measured? How valid are those measurements for in-situ properties? This should be better discussed.

Please note that we are very cautious in describing the criteria and mechanical

parameters chosen in the approach. We recognize that the strength parametrization is the main limitation of our approach. We bring this point carefully in the discussion. Given that we do not have "access to direct strength measurements since no cores were collected" (line 549), our results are discussed at the light of the uncertainty on the strength parameters, as stated line 564 or in the discussion.

Please also note the supplement to this comment:
https://www.solid-earth-discuss.net/se-2019-72/se-2019-72-AC1-supplement.pdf

———————————————————

---

## Referee Comment (RC2) · Anonymous Referee #2 · 30 Apr 2019

The authors present a detailed study in orientation and magnitude of the local stress field at the geothermal site of Rittershoffen in France, near the well-known site Soultz-sous-Forêt. The manuscript focuses on the temporal evolution of borehole breakouts and drilling induced tension fractures using acoustic images of two boreholes acquired by Ultrasonic Borehole Imager in 2012, 2013 and 2105. The manuscript is interesting and provides an important contribution for the understanding of the time-dependent deformation. In this form the manuscript is not ready for publication. Please see my comments.

Major comments: 1. The author mentioned in the abstract that they used for their

investigation image datasets from two boreholes GRT-1 and GRT-2. In the manuscript the analysis as well as the description and the discussion of the results are mainly focused only on GRT-1. I suggest the authors to show only the analysis on GRT-1 well. In case the author want to continue keep also the GRT-2 and detailed analysis of the datasets of this borehole is requested. The analysis must be related to the inclined borehole taking into account the orientation of the principal stresses in an inclined borehole. 2. The authors show in Figure 15 the magnitude of Sv, Sh and SH from 2000 m to 2500 m of GRT-1. To calculate the Sv magnitude the authors used equation (6). The Sv curve is presented as if it were made using a fixed value of 2440 kg/m3 for the entire well. Can you explain why? At 2300 m using equation (6) as the author wrote the value Sv is 54.37 MPa, but using the value of 2570 kg/m3 (Table 3) corresponding to the granite rock at a depth of 2200 m Sv is 58.28 MPa. I suggest redrawing figure 15 showing the entire section of the GRT-1 between 0 and 2562 m (TVD). Furthermore in line 387 the authors should specify that the density value shown in equation (6) is related to the Jurassic rocks between 1172 and 1447 m of GRT-1 as an example, but that the Sv was calculated taking into account the density values of the different rocks at different depths. No Figure for GRT-2. If the authors want to include this well they have to show the data and results.

Minor comments: 1. I suggest that the figures and tables have the same MD or TVD depths, or that both are reported. For example, Table 3 shows lithologies and densities relative to TVD depths, while if I look at the stratigraphy in Figure 8, the lithologies refer to MD depths. 2. Please include also the fractures distribution (number, dip, dip azimuth) highlighting the main faults or fracture zone to better understand the borehole breakout rotation and7or deviation from the mean of Sh. 3. The value from hydraulic test at GRT-1 differs from the data from the boreholes GPK1. Could you explain better the reason? Please add also this Sh- value from GRT-1 in Figure 15 4. The caption of figure 13 refers to figure 12. Whereas the caption of figure 12 refers to figure 13. Please modify. 5. Line 16 GRT-2 instead of GRT2 6. Line 16 2500 m instead of 2500m 7. Line 40 provide an indirect information instead of provide a indirect information 8. Line 90 WSM released in 2016 no in 2008. Please update the reference and cite as: Heidbach, Oliver; Rajabi, Mojtaba; Reiter, Karsten; Ziegler, Moritz; WSM Team (2016): World Stress Map Database Release 2016. GFZ Data Services. http://doi.org/10.5880/WSM.2016.001 (http://dataservices.gfz-potsdam.de/wsm/showshort.php?id=escidoc:1680890) 9. Line 120 grT-1 instead of GRT 1 10. Lines 142-143: please specify which failure condition 11. Lines 307-309 Please insert one or more figures to confirm what has been said. 12. Line 183: why the authors grouped the Triassic sandstone in a single category? Please add in the manuscript the reason: no alteration, homogeneous lithology, no fractures, etc 13. Line 387 Sv [MPa] = 0,024 * z [m] – 0,83 or Sv [MPa] = 0.024 * z [m] – 0.83 but no one value as dot and the other a comma. In order not to confuse the reader, I suggest to use the asterisk (or an x) as a multiplication sign instead of the point. 14. Line 533 please 50 m instead of 50m 15. Line 573 please add a dot after correlation technique 16. Line 579 please add the year of the reservoir stimulation 17. Figure 1: legend: the reference is WSM 2016 not 2006 Helmholtz-Centre Potsdam GFZ. Inset with the sketch of GRT-1 and GRT-2 boreholes: the lithology is not clear, some writings overlap. It would be good if the stratigraphy had the same colours as the geological profile. Highlight the trajectory of the wells on the geological profile. Caption: Heidbach et al., 2016. Cite as: Heidbach, Oliver; Rajabi, Mojtaba; Reiter, Karsten; Ziegler, Moritz; WSM Team (2016): World Stress Map Database Release 2016. GFZ Data Services. 18. Figure 2: Please add two separated scales for radius (mm) and for width (°) 19. Figure 3: show directly in the figure a, b, c, d, the artefacts (signal loss, stick slip). 20. Figure 14: please add the fractures as Tadpole related to this section. 21. Figure 15: Please remove the lithology from inside the figure but add it as litho column to the side of the figure. Please add the fractures as Tadpole related to this section. Is the deviation of the stress values between 2250 and 2380 m, more or less, due to the presence of fractures? 22. Figure 18: Please remove the lithology from inside the figure but add it as litho column to the side of the figure. The symbols of Sh and Sv of GRT-2 are not very clear in the figure. Please change the symbol.

---

## Author Response (AR1)

**Federico Rossetti - Handling Topical Editor**

Solid Earth

Subject: Response to referee comments – manuscript "Stress Characterization and Temporal Evolution of Borehole Failure at the Rittershoffen Geothermal Project" by Jérôme Azzola et al., se-2019-72

**Dear Federico Rossetti,**

We thank you and the referee for the extended reviews of our manuscript. We carefully responded to all the comments and revised the manuscript accordingly. Please find below a point by point response to the major and minor comments of the two reviewers (in black the comments and in blue, our response), followed by a copy of the manuscript with tracked changes. The revised manuscript as well as a copy where the changes made to the original version are highlighted, are attached.

Sincerely, on behalf of the authors Jérôme AZZOLA

**Referee n°1**

This paper addresses the important issue of evaluating a regional stress field from images of two different failure processes (borehole breakouts and so-called drilling Induced fractures) observed in deep boreholes with different orientations, as well as from results from various water injection tests. The methodology is applied at the Rittershoffen site, located 6km east from the Soultz site, where the stress field is quite well known. This is an important contribution for the understanding of stress field in deep rock masses and the quality of images as well as that of their analysis justify completely its publication.

We thank François Cornet for his careful review. We appreciate his recognition of the importance of our contribution.

1. The GRT-2 borehole is inclined  $37^{\circ}$  to the vertical so that the axial and tangential stress components at the borehole wall are not principal stresses. Authors must write down the equations they are considering, including the role of pore pressure, and that of thermal stresses. Indeed, the principal directions, at the wellbore wall, of stresses resulting from the far field stresses are not the same as those of the thermal stresses resulting from the cooling of the rock. This issue is completely ignored, and the paper cannot be published before this is properly dealt with. I encourage authors to look at paper by Wileveau et al. that provides good illustrations of en echelon breakouts observed in inclined wells. (Wileveau Y, F.H. Cornet, J. Desroches and P. Blumling, 2007; Complete in situ stress determination in an argillite sedimentary formation; Physics and Chemistry of the Earth (vol. 32, pp 866-878)

The GRT-2 well is strongly inclined: the mean deviation in the section of interest is  $37^{\circ}$  (as presented in section 2). Equations that describe stress concentration at the borehole wall of a vertical borehole, used for the well GRT-1, are no longer applicable in this case. For the deviated well GRT-2, we used a solution that takes into account the inclined geometry of the borehole (as introduced in section 4.1). The solution is based on equations in which are involved the non-vertical geometry of the well, the orientation of the far field stresses, the thermal stresses and the fluid pressure. We refer to the review of Schmitt et al. (2012) who propose a complete development of the equations, in the general case.

As suggested by the referee, we detailed the used equations and included the computation steps leading to the expression of the effective principal stresses at the borehole wall of the deviated well in the revised version of the manuscript in appendix A. We cite the work of Wileveau et al., as an additional reference to this approach.

2. For their analysis of the width of borehole breakouts, authors refer to three different failure criteria, including the Hoek and Brown criterion. For the parameters to be considered in these criteria, they refer to laboratory work quoted by Rummel, 1991 and by Valley and Evans, 2006. They should also look at the publication by Villeneuve et al. (Villeneuve M.C., M.J. Heap, A.R.L. Kushnir, T. Qin, P. Baud, G. Zhou, and T. Xu, 2018; Estimating in situ rock mass strength and elastic modulus of granite from the Soultz-sous-Forêts geothermal reservoir (France); Geothermal Energy, 6(11), https://doi.org/10.1186/s40517-018-0096-1), which address precisely this issue.

We used all available published data to parametrize our failure criteria, including data provided in Villeneuve et al., 2018, but also from Heap et al. (2019) (which was already cited on lines 90 and 511 of our original manuscript). To clarify this, we added the relevant references in section 5 of our revised manuscript.

3. In their table 3 the density value for the granite is said to be 2570 kg/m3, yet in equation (6) the vertical stress is assumed to be equal to 0.024 z-0.83. These differences should be discussed. In addition, given the vertical stress magnitude is taken into consideration in the three-dimensional failure criteria, authors should show how they determine uncertainties on the vertical stress component evaluation.

The magnitude of the vertical stress Sv is obtained from the weight of the overburden, by integrating the density profile from surface to reservoir depth. We apologize for the typo in Eq. (6), which should read 0.0248 z - 0.83. This misleading rounding will be corrected in the revised manuscript, which leads to a trend in line with density value 2570 kg/m3 chosen for the granitic zone. Given the fact that the vertical stress is obtained by integrating the density profile from surface to reservoir depth, the uncertainty on density add up and thus the uncertainty on the vertical stress estimation increase with depth. Considering an uncertainty of 50 kg/m3 on the densities leads to a 2.5 MPa uncertainty on the vertical stress at reservoir depth. This uncertainty is not significant compared to other uncertainties involved in the analysis as for example those related to the mechanical parameters chosen in the inversion of the maximum horizontal stress.

Details about the uncertainty estimation are added in the revised manuscript, in section 8.1.

**4. Similarly, equations used for the evaluation of the minimum principal stress magnitude is not described and this should be corrected. Evaluation of associated uncertainty should be discussed.**

We follow approaches used in the literature (e.g. Cornet et al., 2007) and estimate the minimum horizontal stress Sh from pressure limiting behavior during hydraulic injections. Since we did not have enough information related to the Rittershoffen site to compute a complete Sh stress profile, we use measurements carried out at the nearby Soultz-sous-Forêts site. The trend is evaluated following Cornet et al. (2007). In their publication, the uncertainty about the trend is largely discussed but not quantified. Deriving confidence bound through data fits or propagating pressure measurement errors leads uncertainty on Sh magnitude of a few megapascals which is irrelevantly low compared to possible misinterpretation of the pressure limiting controlling factors. It is even more insignificant compared to the uncertainty related to the parameterization of the failure criteria that dominates the uncertainty on SH magnitude. We rather complete pragmatically our study by discussing the applicability of the trend to the Rittershoffen site. For this purpose, we analyze the wellhead pressure measured during the hydraulic stimulation of GRT-1 and derived an estimate of Sh at depth from the pressure reached at a maximum flow rate. However, as the pressure shows a gradual but not definitive stabilization for these maximum flow rates, our measurement is discussed as a lower bound of the minimum horizontal stress Sh at depth. We show that our measurement is still consistent with the trend measured at Soultz.

As the second referee also asked for more information on the methodology that we follow, we added details in the manuscript to clarify the steps followed in the estimation of the *Sh* profile.

**5. Table 2 indicates values for the Poisson's ratio, but no reference is made to Young's moduli nor to thermal expansion coefficients used in equation**

We apologize for not having included the Young's moduli and the thermal expansion coefficient values in the manuscript. They have been added to the revised version. The volumetric thermal expansion coefficient is chosen to be constant for the different layers of our model,  $\alpha = 15 \times 10^{-6} \text{ K}^{-1}$ , and the Young's moduli are added in Table 2.

**6. In equation (2) the stress component $\tau$ oct implies the three principal stress components. This should also apply to the mean stress, as opposed to equation written on line 179.**

The original derivation of the equations is proposed by Zimmerman & Al-Ajmi (2006). In their review, authors refer to an "effective mean stress",  $\sigma_{m,2} = \frac{\sigma_1 + \sigma_3}{2}$ , for the Mogi-Coulomb criterion. This is not strictly speaking the mean stress, which would also include a contribution of the intermediate stress  $\sigma_2$ . We clarified the terminology and nomenclature in the revised version of the manuscript.

7. In their discussion of results, authors argue that some of the results obtained for the magnitude of the maximum principal stress magnitude do not satisfy the Coulomb stability condition for the rock mass. Interestingly, Cornet (2016) has argued that the large-scale fluid injections conducted at Soultz have generated large scale failure zones

that are changing in orientation with depth, a feature consistent with the Hoek and Brown criterion but not with a Coulomb criterion. This issue should be discussed more carefully (Cornet, F.H., 2016. Seismic and aseismic motions generated by fluid injections; Geomech. Ener. Env., 5, pp 42-54).

We thank the referee for his comment and suggestion to discuss our results at large scale. It addresses an important issue: the upscaling from the borehole scale (centimeter scale) to the reservoir scale (kilometric scale). Our work is based on the stability of the wellbore wall, i.e. processes that are occurring at a centimetric scale. In our study, we use stability criteria and parameters that have been obtained from laboratory experiments at a similar scale.

On the contrary, Cornet (2016) discusses the strength criterion at the scale of rock mass. To discuss our results in terms of stress profiles at larger scale, we can analyze Fig. 11 which shows the evolution with depth of *SH* over 650m. We see a significant trend of a rotation with depth of the *SH* direction which could be related to the observation of Cornet (2016) and support a Hoek-Brown criterion for failure rather than the Mogi-Coulomb criterion. As discussed in section 9.3, we believe that this trend is rather related to the distance to the fault than to the depth and the effect of the fluid pressure on the Hoek and Brown criterion. From the stability criterion computation, we point out that both the Hoek-Brown and Mogi-Coulomb criteria exceed the frictional limit and cannot be used to choose the most relevant criterion. We should also point out that the injection in Rittershoffen was not as "massive" as in Soultz, i.e. the injected volumes and applied well head pressures were much smaller. The transposition of the knowledge from Soultz in this regard may not be directly applicable.

As requested by the reviewer we developed the discussion on this topic in sections 9.2 and 9.3.

**8. How are the various parameters measured? How valid are those measurements for in-situ properties? This should be better discussed.**

Please note that we are very cautious in describing the criteria and mechanical parameters chosen in the approach. We recognize that the strength parametrization is the main limitation of our approach. We bring this point carefully in the discussion. Given that we do not have "access to direct strength measurements since no cores were collected" (line 582), our results are discussed in the light of the uncertainty on the strength parameters, as stated in section 9.4. In addition to the lack of information to parametrize our criteria, we recognize two other sources of uncertainty:

1) there is no consensus regarding the appropriate failure criterion to asses wellbore wall strength and to be used for borehole breakouts analysis, as mentioned in the manuscript in lines 165-168 (of the original manuscript). In our approach, we used thus multiple criteria and discussed the relevance of the measurements in terms of stress profiles by confronting them to the stability of the rock mass at larger scale.

2) the mechanical and strength parameters that have been selected from core or cutting analyses are not necessarily representative of the *in-situ* conditions.

We added new elements of the discussion into the revised manuscript in section 5 and in section 9.3.

**9. Caption of figure 12 has been exchanged with that of fig 13.**

We thank the reviewer for pointing out the caption swap between fig. 12 and 13. We fixed this issue in the revised manuscript.

**Referee n°2**

The authors present a detailed study in orientation and magnitude of the local stress field at the geothermal site of Rittershoffen in France, near the well-known site Soultz-sous-Forêts. The manuscript focuses on the temporal evolution of borehole breakouts and drilling induced tension fractures using acoustic images of two boreholes acquired by Ultrasonic Borehole Imager in 2012, 2013 and 2105. The manuscript is interesting and provides an important contribution for the understanding of the time-dependent deformation.

In this form the manuscript is not ready for publication. Please see my comments.

We thank the reviewer for his careful review. We appreciate his interest in the manuscript and his recognition of the importance of our contribution.

**Major comments:**

1. The author mentioned in the abstract that they used for their investigation image datasets from two boreholes GRT-1 and GRT-2. In the manuscript the analysis as well as the description and the discussion of the results are mainly focused only on GRT-1. I suggest the authors to show only the analysis on GRT-1 well. In case the author wants to continue keep also the GRT-2, a detailed analysis of the datasets of this borehole is requested. The analysis must be related to the inclined borehole taking into account the orientation of the principal stresses in an inclined borehole.

We acknowledge that the description and the discussion of the results are mainly focused on the data of the GRT-1 well, as the quality of data from GRT-2 is generally lower than for GRT-1 (line 235). The image quality problems with GRT-2 are detailed in section 6.1 of the manuscript and illustrated in figure 3.c. It shows in particular the significant stick-slip effect inducing alternative compression and stretching of the UBI images. Figure 3.d. is an example of an erroneous borehole radius record. Given the extent of the artefacts highlighted in GRT-2, the measurements of the breakout parameters in this borehole are more uncertain than in GRT-1 and no DIFTS have been measured in GRT-2 (line 325). We still analysed the stress tensor in GRT-2 using a proper deviated well approach, which has been clarified in appendix A of the revised manuscript. We feel that it is worth adding the GRT-2 data in the manuscript as the expression of the measurements in TVD enables to compare the results with the measurements performed in GRT-1, even if the data quality doesn't enable to propose an extended analysis of the stress tensor as in GRT-1.

2. The authors show in Figure 15 the magnitude of Sv, Sh and SH from 2000 m to 2500 m of GRT-1. To calculate the Sv magnitude the authors used equation (6). The Sv curve is presented as if it were made using a fixed value of 2440 kg/m3 for the entire well. Can you explain why? At 2300 m using equation (6) as the author wrote the value Sv is 54.37 MPa but using the value of 2570 kg/m3 (Table 3) corresponding to the granite rock at a depth of 2200 m Sv is 58.28 MPa.

The magnitude of the vertical stress Sv is obtained from the weight of the overburden. The density profile provided in Table 3, is integrated from surface to maximum depth. The trend provided in equation (6) is obtained from a linear fit to the measurements in the range of depths considered in our study. We apologize for the typo in Eq. (6), which should read  $0.0248 \ z - 0.83$ .

This misleading rounding is corrected in the revised manuscript, which leads to a trend in line with density value 2570 kg/m3 chosen for the granitic layer.

**I suggest redrawing figure 15 showing the entire section of the GRT-1 between 0 and 2562 m (TVD).**

We acknowledge and tested this advice, but by redrawing figure 15 from 0 to 2562 m, we considerably deteriorate the readability of the measurements presented in the figure 15 for greater depths, from 1950 to 2550m, while showing a long wellbore section (from 0 to 1950 m) without data.

After careful consideration, we decided thus not to extend the vertical scale to the entire GRT-1 section.

Furthermore, in line 387 the authors should specify that the density value shown in equation (6) is related to the Jurassic rocks between 1172 and 1447 m of GRT-1 as an example, but that the Sv was calculated taking into account the density values of the different rocks at different depths. No Figure for GRT-2. If the authors want to include this well, they have to show the data and results.

We added details about the procedure followed (after line 394). We measure Sv as a function of TVD in order to apply the same measurements in both wells. Figure 17 and 18 are showing results for GRT-2.

**Minor comments:**

1. I suggest that the figures and tables have the same MD or TVD depths, or that both are reported. For example, Table 3 shows lithologies and densities relative to TVD depths, while if I look at the stratigraphy in Figure 8, the lithologies refer to MD depths.

TVD is the most relevant depth scale to present stress estimate and to compare results across both wells, with GRT-1 almost vertical (TVD and MD are not very different) and GRT-2 being deviated.

We follow thus the advice of the reviewer and made sure to add a TVD depth scale on all our figures and tables.

**2. Please include also the fractures distribution (number, dip, dip azimuth) highlighting the main faults or fracture zone to better understand the borehole breakout rotation and/or deviation from the mean of *Sh*.**

The major fracture network was observed from acoustic wall imagery in the open-hole sections of GRT-1 and GRT-2 by Vidal (2017). Major continuous fractures (thickness measured on acoustic images higher than 1 cm) are analyzed in both wells. The detailed structural survey is available in Appendix 2 of Vidal's thesis. The fractures are oriented globally in GRT-1 N 15° E to N 20° E with a dip of 80° W. In GRT-2, the main fracture family is oriented N 155° E to N 175° E with a dip of 80° E to 90° E. Fracture density is highest on the roof of the granitic basement. These summary elements are added in section 2 of the revised manuscript, which details the context of the Rittershoffen project. Our analysis doesn't consist in the measurement / discussion of the distribution and orientation of the natural fractures highlighted through the GRT-1 and GRT-2 wells, which has been extensively studied by Jeanne Vidal in her thesis.

We believe thus that adding data regarding the fracture distribution and orientation in the figures doesn't contribute to the discussion of the proposed measurements but would necessitate to analyse data that are not in the focus of our paper.

**3. The value from hydraulic test at GRT-1 differs from the data from the boreholes GPK1. Could you explain better the reason? Please add also this Sh- value from GRT-1 in Figure 15**

The approach followed has certainly not been sufficiently clearly explained, and we added details in the section 8.3 to carefully describe the steps in the estimation of the *Sh* profile proposed in the GRT-1 and GRT-2 wells.

The profile of the minimum horizontal stress Sh is estimated from pressure limiting behavior during hydraulic injections. Since we did not have enough information related to the Rittershoffen project to compute a complete Sh stress profile, we used measurements carried out at the nearby Soultz-sous-Forêts project. The trend is evaluated after Cornet et al. (2007). To complete our analysis, we analyzed the wellhead pressure measured during the hydraulic stimulation of GRT-1 and derived an "estimate at best" of the Sh magnitude at depth from the pressure reached at maximum flow rate. The wellhead pressure measured at 1913m in GRT-1 during the hydraulic stimulation (data provided in figure fig. 12) shows a gradual but not definitive stabilization at flow rates up to 80 L.s-1. Even if the pressure limiting behavior, related to the creation or reactivation of faults, is not reached, we discuss the measurement as a lower bound for the minimum horizontal stress Sh at 1913m. By comparing our measurement in Rittershoffen at 1913m with the trend considered in the stress analysis and measured originally in Soultz-sous-Forêts, we show that both measurements are consistent and that the Rittershoffen measurement is indeed a realistic lower bound for the chosen trend.

**4. The caption of figure 13 refers to figure 12. Whereas the caption of figure 12 refers to figure 13. Please modify.**

The caption of figure 13 has been inverted with caption of figure 12, as mentioned by both referees. We fixed this issue in the revised manuscript.

5. Line 16 GRT-2 instead of GRT2

6. Line 16 2500 m instead of 2500m

7. Line 40 provide an indirect information instead of provide a indirect information

8. Line 90 WSM released in 2016 no in 2008. Please update the reference and cite as: Heidbach, Oliver; Rajabi, Mojtaba; Reiter, Karsten; Ziegler, Moritz; WSM Team (2016): World Stress MapDatabase Release 2016. GFZ Data Services. http://doi.org/10.5880/WSM.2016.001

(http://dataservices.gfz-potsdam.de/wsm/showshort.php?id=escidoc:1680890)

**9. Line 120 GRT-1 instead of GRT 1**

We corrected the typographical errors referenced in comment #5, #6, #7 and #9 in the revised manuscript. We updated the reference to the WSM (comment #8).

**10. Lines 142-143: please specify which failure condition**

Details have been added to the manuscript regarding the failure criterion used to the above-mentioned lines.

11. Lines 307-309 Please insert one or more figures to confirm what has been said.

The request of the referee is not very clear, as the lines referred to do not highlight an obvious lack in information. If the referee refers to the deviation of the wells in the open-hole section, this is shown in Figure 1: the trajectories of GRT-1 and GRT-2 show that the deviations are constant in the section of interest and that GRT-1 is quasi-vertical.

**12. Line 183: why the authors grouped the Triassic sandstone in a single category? Please add in the manuscript the reason: no alteration, homogeneous lithology, no fractures, etc**

The sandstones crossed by the open section of the well are all from the Buntsandstein (section 5 of the manuscript). Heap et al, (2019) studied in detail the strength evolution with depth of the Buntsandstein mechanical properties. As suggested by the referee, they evidenced significant variations of the compressive strength together with elastic modulus changes. They also pointed out the role of the fluid content on the UCS. However, these variations are limited compared to the statistical fluctuations of our measurement. Accordingly, we gathered the Buntsandstein sandstones as a single unit (after line 203). We used typical strength parameters from Hoek and Brown (1997) to characterize the geological unit.

13. Line 387 Sv [MPa] = 0,024 \* z [m] - 0,83 or Sv [MPa] = 0.024 \* z [m] - 0.83 but no one value as dot and the other a comma. In order not to confuse the reader, I suggest using the asterisk (or an x) as a multiplication sign instead of the point.

To avoid any confusion for the reader about the punctuation used in the equations, we followed the referee suggestions and replaced the dot by an asterisk in the equations proposed in the manuscript.

- 14. Line 533 please 50 m instead of 50m
- 15. Line 573 please add a dot after correlation technique
- 16. Line 579 please add the year of the reservoir stimulation

We modified typographical errors previously referenced (comment #14 and #15) and added details to the manuscript regarding the year of the stimulation to the above-mentioned lines (comment #16).

17. Figure 1: legend: the reference is WSM 2016 not 2006 Helmholtz-Centre Potsdam GFZ. Inset with the sketch of GRT-1 and GRT-2 boreholes: the lithology is not clear, some writings overlap. It would be good if the stratigraphy had the same colours as the geological profile. Highlight the trajectory of the wells on the geological profile. Caption: Heidbach et al., 2016. Cite as: Heidbach, Oliver; Rajabi, Mojtaba; Reiter, Karsten; Ziegler, Moritz; WSM Team (2016): World Stress Map Database Release 2016. GFZ Data Services.

The legend has been updated as well as the caption with the reference proposed by the referee. The writings in the lithology and in the legend have been made clearer. The lithological profile has been set in agreement with the stratigraphy (bottom and left inserts). The geological profile includes the trajectory of the wells even if its scale doesn't enable to distinguish the direction of GRT-1 and GRT-2.

18. Figure 2: Please add two separated scales for radius (mm) and for width (\_)

Figure 2 has been updated accordingly to the referee's suggestions.

19. Figure 3: show directly in the figure a, b, c, d, the artefacts (signal loss, stick slip).

In order not to load unnecessarily the figure, we added details in the figure caption regarding to the artefacts. We feel that the mentioned artefacts are now easily recognizable in the images of Fig. 3.

20. Figure 14: please add the fractures as Tadpole related to this section.

In our analysis, we didn't study the distribution and orientation of the fractures, which has been done by Jeanne Vidal in her thesis (Vidal, 2017). We believe that adding data regarding the fracture distribution and orientation in our figures doesn't contribute to the discussion of the proposed measurements but would necessitate to analyse data that are not in the focus of our paper.

21. Figure 15: Please remove the lithology from inside the figure but add it as litho column to the side of the figure. Please add the fractures as Tadpole related to this section. Is the deviation of the stress values between 2250 and 2380 m, more or less, due to the presence of fractures?

We removed the lithology from the inside of the figure and added it to the side of the figure. The deviation of the stress values is correlated to the increase in the breakout width at the mentioned depths.

22. Figure 18: Please remove the lithology from inside the figure but add it as litho column to the side of the figure. The symbols of Sh and Sv of GRT-2 are not very clear in the figure. Please change the symbol.

We removed the lithology from the inside of the figure and added it to the side of the figure. We modified the symbols related to the stress state estimates in GRT-2 to improve readability.

[revised manuscript text omitted]
 - Pf + \sigma^{\Delta T}_{\theta} $ (A3)                                                                                                                                                                                                                                                                                                                                                                                                                                                                                                                                                                                                                                                                                                                                        |
| 713               | $\underline{\sigma_{\zeta\zeta}} = \beta 1 - 4 \nu \left( \alpha 2 \cos 2\theta + \alpha 3 \sin 2\theta \right) \tag{A4}$                                                                                                                                                                                                                                                                                                                                                                                                                                                                                                                                                                                                                                                                                                                                                           |
| 714               | $\underline{\tau}_{\theta\zeta} = 2 \gamma 1 \cos \theta + 2 \gamma 2 \sin \theta \tag{A5}$                                                                                                                                                                                                                                                                                                                                                                                                                                                                                                                                                                                                                                                                                                                                                                                         |
| 715               | $\underline{\tau}_{\underline{r}\zeta} = 0 \tag{A6}$                                                                                                                                                                                                                                                                                                                                                                                                                                                                                                                                                                                                                                                                                                                                                                                                                                |
| 716               | $\underline{\tau_{\theta r}} = 0 \tag{A7}$                                                                                                                                                                                                                                                                                                                                                                                                                                                                                                                                                                                                                                                                                                                                                                                                                                          |
| 717               |                                                                                                                                                                                                                                                                                                                                                                                                                                                                                                                                                                                                                                                                                                                                                                                                                                                                                     |
| 718               | The geometrical coefficients involved in Eqs. (A2-A7) are expressed as a function of the three far-field principal stress state                                                                                                                                                                                                                                                                                                                                                                                                                                                                                                                                                                                                                                                                                                                                                     |
| 719               | $[\sigma_{x'x'}, \sigma_{y'y'}, \sigma_{z'z'}]$ and as a function of the geometrical rotations aij :                                                                                                                                                                                                                                                                                                                                                                                                                                                                                                                                                                                                                                                                                                                                                                         |
| 720               |                                                                                                                                                                                                                                                                                                                                                                                                                                                                                                                                                                                                                                                                                                                                                                                                                                                                                     |
| 721               | $\frac{\alpha 1 = \frac{1}{2} \left[ \left( \frac{a_{x'x}}{a_{x'x}} \sin^2 \Phi + \frac{a_{x'y}}{a_{x'z}} + \frac{a_{x'z}}{a_{x'z}} \cos^2 \Phi - 2 \frac{a_{x'z}}{a_{x'z}} \sin \Phi \cos \Phi \right) \sigma_{x'x'} + \left( \frac{a_{y'x}}{a_{y'x}} \sin^2 \Phi + \frac{a_{y'y}}{a_{y'z}} + \frac{a_{y'z}}{a_{y'z}} \cos^2 \Phi - 2 \frac{a_{y'z}}{a_{y'z}} \sin \Phi \cos \Phi \right) \sigma_{x'x'} + \left( \frac{a_{y'x}}{a_{y'x}} \sin^2 \Phi + \frac{a_{y'y}}{a_{y'z}} + \frac{a_{y'z}}{a_{y'z}} \cos^2 \Phi - 2 \frac{a_{y'z}}{a_{y'z}} \sin \Phi \cos \Phi \right) \sigma_{x'x'}$                                                                                                                                                                                                                                                                                        |
| 722               | $\underline{\cos \Phi}  \sigma_{y'y'} + (  a^2_{z'x} \sin^2 \Phi + a^2_{z'y} + a^2_{z'z} \cos^2 \Phi - 2  a^2_{z'z}  a^2_{z'x} \sin \Phi  \cos \Phi)  \sigma_{z'z'}  ] \tag{A8}$                                                                                                                                                                                                                                                                                                                                                                                                                                                                                                                                                                                                                                                                                                    |
| 723               | $\underline{\alpha 2} = \frac{1}{2} \left[ \left( -\frac{a^2_{x'x}}{\sin^2 \Phi} + \frac{a^2_{x'y}}{a^2_{x'z}} \cos^2 \Phi + 2 \frac{a^2_{x'z}}{a^2_{x'x}} \sin \Phi \cos \Phi \right) \sigma_{x'x'} + \left( -\frac{a^2_{y'x}}{\sin^2 \Phi} + \frac{a^2_{y'y}}{a^2_{y'z}} \cos^2 \Phi + 2 \frac{a^2_{y'z}}{a^2_{y'x}} \sin \Phi \cos \Phi \right) \sigma_{x'x'} + \left( -\frac{a^2_{y'x}}{a^2_{y'x}} \sin^2 \Phi + \frac{a^2_{y'z}}{a^2_{y'z}} \cos^2 \Phi + 2 \frac{a^2_{y'z}}{a^2_{y'x}} \sin \Phi \cos \Phi \right) \sigma_{x'x'} + \left( -\frac{a^2_{y'x}}{a^2_{y'x}} \sin^2 \Phi + \frac{a^2_{y'z}}{a^2_{y'z}} \cos^2 \Phi + 2 \frac{a^2_{y'z}}{a^2_{y'x}} \sin \Phi \cos \Phi \right) \sigma_{x'x'} + \left( -\frac{a^2_{y'x}}{a^2_{y'x}} \sin^2 \Phi + \frac{a^2_{y'z}}{a^2_{y'x}} \cos^2 \Phi + 2 \frac{a^2_{y'z}}{a^2_{y'x}} \sin \Phi \cos \Phi \right) \sigma_{x'x'}$ |
| 724               | $\underline{\cos \Phi}  \sigma_{y'y'} + (-a_{z'x}^2 \sin^2 \Phi + a_{z'y}^2 - a_{z'z}^2 \cos^2 \Phi + 2  a_{z'z}^2  a_{z'x}^2 \sin \Phi \cos \Phi)  \sigma_{z'z'} ] \tag{A9}$                                                                                                                                                                                                                                                                                                                                                                                                                                                                                                                                                                                                                                                                                                       |
| 725               | $\underline{\alpha 3} = (\underline{a_{x'y}}, \underline{a_{x'z}}\cos \Phi - \underline{a_{x'x}}, \underline{a_{x'y}}\sin \Phi) \sigma_{x'x'} + (\underline{a_{y'y}}, \underline{a_{y'z}}\cos \Phi - \underline{a_{y'x}}, \underline{a_{y'y}}\sin \Phi) \sigma_{y'y'} + (\underline{a_{z'y}}, \underline{a_{z'z}}\cos \Phi - \underline{a_{z'x}}, \underline{a_{z'y}}\sin \Phi) \sigma_{z'z'}$                                                                                                                                                                                                                                                                                                                                                                                                                                                                                      |
| 726               | (A10)                                                                                                                                                                                                                                                                                                                                                                                                                                                                                                                                                                                                                                                                                                                                                                                                                                                                               |

| $728  \underline{-\sin^2\Phi} ]  \underline{\sigma_{y'y'}} + [\underline{-a^2_{z'x}}\sin\Phi\cos\Phi + \underline{a^2_{z'z}}\cos\Phi\sin\Phi + \underline{a_{z'z}}\underline{a_{z'x}}(\cos^2\Phi - \sin^2\Phi)] \\ \underline{\sigma_{z'z'}} ] \tag{
[revised manuscript text omitted]

---

## Author Response (AR2)

Federico Rossetti - Handling Topical Editor

Solid Earth

June 18, 2019

Subject: Response to referee comments – manuscript "Stress Characterization and Temporal Evolution of Borehole Failure at the Rittershoffen Geothermal Project" by Jérôme Azzola et al., se-2019-72

Dear Federico Rossetti,

We thank you and the referee for the acceptance of our manuscript for publication in Solid Earth. We responded to the comment of the referee and revised the manuscript accordingly. Please find below a response to the comment (in black the comments and in blue, our response), followed by a copy of the manuscript with tracked changes.

Sincerely, on behalf of the authors
Jérôme AZZOLA
* * *
I have read with great interest this revised version of the manuscript and I was pleased to note that appendix A provides now the equations considered for describing the stress field in the vicinity of the inclined well.

In particular, equations A6 and A7 are valid only for certain orientations of the well GRT-2 with respect to the principal stress direction. Indeed, in the most general case, these terms are not zeros. But I was not able to find in this revised version the direction of well GRT-2 in its deviated part, so that I cannot verify that indeed the hypothesis implied by equations A6 and A7 are validated by field observations. So, I do recommend that authors provide at some point the direction of the deviated part of well GRT-2.

I may point out that the fact that break-outs are collinear with the borehole axis for GRT-1 demonstrates that indeed, the vertical direction is principal. Had the well been inclined by more than 20° to any of the principal stress directions, both the DITF and the break-outs would have been "en-échelon", as was observed e.g. by Wileveau et al., quoted by authors, in highly deviated wells. These en-échelon patterns are observed when the inclination of the well with respect to any of the principal stress directions reaches more than 20°.

A small discussion of this issue would have been welcome. Nevertheless, if authors are happy with this version of the paper, I am happy to let the paper go as is…

We thank François Cornet for his comment. The direction of the deviated part of well GRT-2, which is north directed, was given line 495-496 of the revised manuscript. However, we added the information in Figure 1 (lower left insert) and in the section 2 of the manuscript (line 74). In addition, we point out that the drilling direction is therefore close to the direction of one of the principal stress, with a difference of less than 20° (lines 96-97). We discuss briefly the implications on the assessment of the principal stress directions from line 139 to 142.

[revised manuscript text omitted]